

# The diverse radiodont fauna from the Marjum Formation of Utah, USA (Cambrian: Drumian)

Stephen Pates[1],[*], Rudy Lerosey-Aubril[1],[*], Allison C. Daley[2], Carlo Kier[3], Enrico Bonino[3] and Javier Ortega-Hernández[1]

[1] Museum of Comparative Zoology and Department of Organismic and Evolutionary Biology, Harvard University, Cambridge, MA, USA
[2] Institut des sciences de la Terre (ISTE), Université de Lausanne, Lausanne, Vaud, Switzerland
[3] Back to the Past Museum, Carretera Cancún, Quintana Roo, Mexico
[*] These authors contributed equally to this work.

Corresponding authors
Stephen Pates,
stephenpates@fas.harvard.edu
Rudy Lerosey-Aubril,
rudy_lerosey@fas.harvard.edu

## ABSTRACT

Radiodonts have long been known from Cambrian deposits preserving non-biomineralizing organisms. In Utah, the presence of these panarthropods in the Spence and Wheeler (House Range and Drum Mountains) biotas is now well-documented. Conversely, radiodont occurrences in the Marjum Formation have remained scarce. Despite the large amount of work undertaken on its diverse fauna, only one radiodont (*Peytoia*) has been reported from the Marjum Biota. In this contribution we quadruple the known radiodont diversity of the Marjum fauna, with the description of the youngest members of two genera, *Caryosyntrips* and *Pahvantia*, and that of a new taxon *Buccaspinea cooperi* gen. et sp. nov. This new taxon can be identified from its large oral cone bearing robust hooked teeth with one, two, or three cusps, and by the unique endite morphology and organisation of its frontal appendages. Appendages of at least 12 podomeres bear six recurved plate-like endites proximal to up to four spiniform distal endites. *Pahvantia hastata* specimens from the Marjum Formation are particularly large, but otherwise morphologically indistinguishable from the carapace elements of this species found in the Wheeler Formation. One of the two new *Caryosyntrips* specimens can be confidently assigned to *C. camurus*. The other bears the largest spines relative to appendage length recorded for this genus, and possesses endites of variable size and unequal spacing, making its taxonomic assignment uncertain. *Caryosyntrips, Pahvantia*, and *Peytoia* are all known from the underlying Wheeler Formation, whereas isolated appendages from the Spence Shale and the Wheeler Formation, previously assigned to *Hurdia*, are tentatively reidentified as *Buccaspinea*. Notably, none of these four genera occurs in the overlying Weeks Formation, providing supporting evidence of a faunal restructuring around the Drumian-Guzhangian boundary. The description of three additional nektonic taxa from the Marjum Formation further documents the higher relative proportion of free-swimming species in this biota compared to those of the Wheeler and Weeks Lagerstätten. This could be related to a moderate deepening of the basin and/or changing regional ocean circulation at this time.

## INTRODUCTION

Fossil deposits that preserve the remains of both biomineralizing and non-biomineralizing organisms provide key insights into the evolution and ecology of life on Earth not accessible from the shelly fossil record alone. Such exceptional strata, or Konservat-Lagerstätten, have been discovered on all major Cambrian palaeocontinents. Most Konservat-Lagerstätten are known from China and North America (*Muscente et al., 2017*), and although these deposits are not equally prolific in terms of taxonomic diversity and fossil abundance (*Gaines, 2014*), suffer from different taphonomic biases (*Saleh et al., 2020*), and vary at both regional and continental scales (*Holmes, García-Bellido & Lee, 2018*; *Fu et al., 2019*; *Nanglu, Caron & Gaines, 2020*), they provide congruent pictures of how bilaterian animals diversified, became ecologically significant, and profoundly influenced marine environments at that time (*Budd & Jensen, 2000*; *Butterfield, 2011*, *2018*; *Erwin & Tweedt, 2012*; *Mángano & Buatois, 2014*, *2020*; *Daley et al., 2018*). The most abundant and diverse group of these early bilaterians were total-group euarthropods, relatives of modern arachnids, crustaceans, insects, and myriapods. Iconic in Paleozoic exceptionally-preserved faunas, stem-group euarthropods (sensu *Ortega-Hernández, 2016*) such as 'gilled-lobopodians' and radiodonts have proved critical for our understanding of the early evolution of the phylum (*Budd, 1998*; *Daley, 2013*; *Daley et al., 2009*, *2018*; *Cong et al., 2014*; *Vannier et al., 2014*; *Van Roy, Daley & Briggs, 2015*; *Young & Vinther, 2017*).

Radiodonts—a diverse extinct group that includes *Anomalocaris* and its relatives—have long been known as comparatively common elements in Cambrian Konservat-Lagerstätten, but an ever-growing body of evidence shows that these organisms occupied a variety of ecological niches and contributed in different ways to the diversity of early animal communities (*Daley & Budd, 2010*; *Daley & Edgecombe, 2014*; *Vinther et al., 2014*; *Van Roy, Daley & Briggs, 2015*; *Lerosey-Aubril & Pates, 2018*; *Liu et al., 2018*; *Moysiuk & Caron, 2019*). Radiodonts greatly differed in size, ranging from millimetres to meters in length (*Van Roy, Daley & Briggs, 2015*; *Lerosey-Aubril & Pates, 2018*; *Liu et al., 2018*; *Pates et al., 2020a*), and had variable body shapes that impacted their swimming capabilities. Radiodonts are typically reconstructed as nektonic animals (also referred to as 'free swimmers' hereafter), with the possible exception of the eudemersal *Cambroraster* (*Moysiuk & Caron, 2019*; *Liu et al., 2020*). Forms with elongate swimming flaps and reduced cephalic sclerites (amplectobeluids and anomalocaridids; *Daley & Edgecombe, 2014*; *Cong et al., 2014*, *2016*, *2017*, *2018*; *Liu et al., 2018*) were likely more agile swimmers than those with comparatively reduced, but paired flaps, and cylindrical bodies made semi-rigid by the presence of an elongate cephalic carapace (e.g. the hurdiids *Aegirocassis* and *Hurdia*; *Daley et al., 2009*; *Daley, Budd & Caron, 2013*; *Van Roy, Daley & Briggs, 2015*). Swimming power has been shown to increase with the size of swimming flaps (*Usami, 2006*), which are more developed in amplectobeluids and anomalocaridids. These two

families of inferred ambush predators also differ from hurdiids by the presence of a large tail fan, a structure that increases manoeuvrability and reduces turning radii, as demonstrated by experimental fluid dynamics (*Sheppard, Rival & Caron, 2018*). Hurdiids may have been adapted for agile swimming at lower speeds, based on the presence of paired body flaps, with the ventral flaps interpreted as being used mainly for propulsion, and the dorsal flaps providing stability and steering particularly during the sustained gliding that likely characterised the swimming motion in suspension feeding taxa (e.g. *Aegirocassis, Pahvantia*) (*Van Roy, Daley & Briggs, 2015*; *Lerosey-Aubril & Pates, 2018*).

This interpretation of radiodonts as ecologically diverse components of early Paleozoic faunas finds additional support in the recent realization that many Cambrian Konservat-Lagerstätten host several representatives of this major group. For instance, radiodonts are represented by at least nine genera in Chengjiang (all localities together; *Zeng et al., 2018*, tab. S1; *Cong et al., 2018*; *Guo et al., 2019*; *Liu et al., 2020*), seven genera in the Burgess Shale (*Moysiuk & Caron, 2019*; *Zeng et al., 2018*, tab. S1)—three to five of them occurring in the most studied localities (*Daley & Budd, 2010*; *Daley, Budd & Caron, 2013*; *O'Brien & Caron, 2016*)—and four genera in the Kinzers Formation (*Pates & Daley, 2019*). Such co-occurrences are made possible by the occupation of different ecological niches, but also reflect patchiness in the preservation of palaeocommunities within these deposits (*Nanglu, Caron & Gaines, 2020*). Recent studies in Utah, the home of five Cambrian Konservat-Lagerstätten (*Robison, Babcock & Gunther, 2015*), have increased the known diversity of the radiodont faunas in the Spence Shale (three genera; *Briggs et al., 2008*; *Pates & Daley, 2017*; *Pates, Daley & Lieberman, 2018*), the Wheeler Formation in the Drum Mountains (three genera; *Halgedahl et al., 2009*, fig. 10L; *Pates & Daley, 2017*; *Lerosey-Aubril & Pates, 2018*), and especially the Wheeler Formation in the House Range (at least six genera; *Briggs et al., 2008*; *Pates, Daley & Ortega-Hernández, 2017*, *2018*; *Lerosey-Aubril & Pates, 2018*; *Pates, Daley & Lieberman, 2018*; *Lerosey-Aubril et al., 2020*). Only one radiodont genus (*Anomalocaris*) has as-yet been reported from the Weeks Formation (*Lerosey-Aubril et al., 2014*), but this is the least explored and the youngest of the Cambrian Konservat-Lagerstätten of Utah (*Lerosey-Aubril et al., 2018*). By contrast, the Marjum Formation has received considerable attention by both professional and amateur palaeontologists over the last 50 years (*Robison, 1991*; *Bonino & Kier, 2010*; *Conway Morris, 2015*; *Robison, Babcock & Gunther, 2015*), and yet only two specimens of a single radiodont genus, *Peytoia*, have been described until now (*Briggs & Robison, 1984*; *Pates, Daley & Lieberman, 2018*). This is all the more surprising as the Marjum fauna is particularly diverse (over 139 species according to *Robison, Babcock & Gunther, 2015*, but see below) with a high proportion of pelagic taxa (more than one third of the generic diversity; see definition of pelagic below).

In this contribution, we report the first occurrences in the Marjum Formation, and the youngest occurrences overall, of the radiodont genera *Caryosyntrips* and *Pahvantia*, alongside the description of the new hurdiid *Buccaspinea cooperi* gen. et sp. nov. Beyond complementing our understanding of these genera, the new data confirm that the Marjum assemblage is proportionally richer in free swimming components than those of the

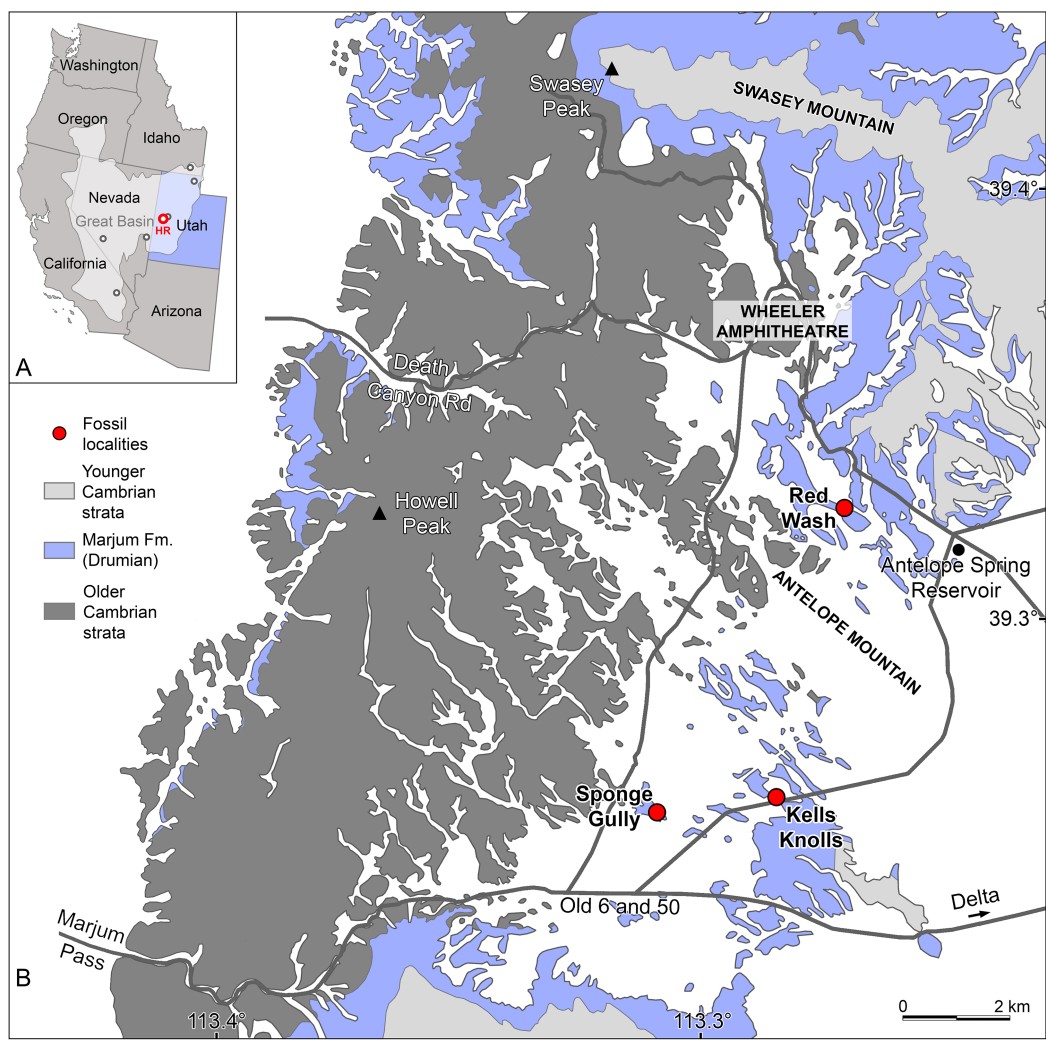

**Figure 1 Radiodont occurrences in the Cambrian (Drumian) Marjum Formation in the House Range of Utah, USA.** (A) Map of western USA showing the locations of the main Cambrian Konservat-Lagerstätten (circles) of the Great Basin (light grey area), including the Marjum Formation in the House Range (HR) of western Utah (credit: Rudy Lerosey-Aubril). (B) Simplified geological map of the central House Range (credit: Rudy Lerosey-Aubril), showing the geographic distribution of the Marjum Formation and the localities that have yielded radiodont fossils: Kells Knolls (*Buccaspinea cooperi* gen. et sp. nov.), Red Wash (*Caryosyntrips* sp.), and Sponge Gully (*Peytoia nathorsti*). A fourth radiodont, *Pahvantia hastata*, occurs in the Marjum Formation, but the known material of this taxon is of uncertain origin within the House Range. Data relating to the spatial distributions of Cambrian strata are derived from *Hintze (1980)*.

underlying Wheeler and overlying Weeks formations, which might be due to a local increase in bathymetry of the depositional environment, enhanced faunal mixing resulting from changes in ocean circulation, or a combination of these factors.

## GEOLOGICAL SETTING

The new radiodont specimens described in this contribution were collected from the Marjum Formation of the House Range of Utah (Fig. 1). This unit was deposited along the northern seaward margin (now western North America) of the 'Great American

Carbonate Bank' (*Derby et al., 2012*) which encircled Laurentia at this time and separated shallow-water proximal shelf settings from deep-water distal shelf and slope environments. The development of extensive carbonate facies all around Laurentia during the early Cambrian was facilitated by its low-latitude palaeogeographic position (*Torsvik & Cocks, 2017*). The Marjum Formation and other Cambrian Lagerstätten from western North America were deposited close to the equator.

The Marjum Formation was deposited within the House Range Embayment, a fault-controlled basin that developed during the Wuliuan age and formed a prominent re-entrant within the offshore margin of the carbonate platform in parts of present-day Nevada and Utah (*Hintze & Robison, 1975*; *Rees, 1986*). This locus of deep-water sedimentation within the Great Carbonate Bank allowed the deposition of a continuous succession of shale-dominated strata typical of the Outer Detrital Belt (sensu *Aitken, 1997*), namely the Wheeler, Marjum, and Weeks Formations (in ascending stratigraphic order). There is clear evidence for the presence of a gently sloping ramp connecting the deepest part of the embayment to the carbonate platform to the east (now north; *Rees, 1986*; *Foster & Gaines, 2016*). In contrast, the transition between the two areas in the west (now south) might have been abrupt (*Rees, 1986*). The Marjum Formation crops out in parts of the House Range of Utah, but not in neighbouring ranges. This more limited geographic extent compared to that of the underlying Wheeler Formation (House Range and Drum Mountains) records a general filling of the basin. Reaching up to ca. 430 m in thickness (*Miller, Evans & Dattilo, 2012*), the Marjum Formation is composed of thin-bedded limestone inter-bedded with shale/lime mudstone (*Robison, 1964*), which have yielded a diverse biota of about 145 species (89 genera), of which 30 are entirely non-biomineralizing organisms (*Robison, Babcock & Gunther, 2015*; this study). This important diversity of the Marjum biota is partially explained by the fact that the unit extends from the Drumian to the Guzhangian through three agnostoid biozones (*Ptychagnostus atavus*, *P. punctuosus*, and *Lejopyge laevigata* biozones; *Robison & Babcock, 2011*). However, to our knowledge non-biomineralized fossils have only been recovered from the middle part of the unit (*P. punctuosus* biozone) and accordingly are all Drumian in age.

## MATERIALS AND METHODS

The material described in this contribution consists of new specimens from the Marjum Formation, which are deposited in the collections of the Back to the Past Museum (prefix BPM) and the Natural History Museum of Utah (prefix UMNH.IP). Photographs of additional fossils are used for comparative purposes and to illustrate pelagic components of the Marjum fauna—these specimens are housed in the Biodiversity Institute of the University of Kansas (prefix KUMIP), the Department of Geology of the University of Utah (prefix UU), the Natural History Museum of Utah, and the Smithsonian Institution's U.S. National Museum of Natural History (prefix USNM-PAL). Lastly, materials accessioned at the Museum of Comparative Zoology, Harvard University (prefix MCZ) were examined for comparative purposes but not figured. Details for all specimens

considered over the course of this study can be found in the Supplementary Data (*Pates et al., 2020b*, table S1).

*Robison, Babcock & Gunther (2015)* comprehensive list of taxa present in the Wheeler (House Range), Marjum, and Weeks Formations was used to create a database to compare the compositions of their exceptionally-preserved faunas with regard to taxonomy and life habits (*Pates et al., 2020b*; Data S3). This dataset was refined using taxonomic lists compiled by R. A. Robison, which detail the compositions of fossil assemblages at most Wheeler, Marjum, and Weeks fossil sites in the House Range of Utah. This enabled us to exclude the taxa that only occur in stratigraphic intervals barren of non-biomineralized fossils, such as the lower and upper parts of the Marjum Formation or the lower part of the Weeks Formation. We then updated the resulting dataset to include omitted (*Caron, Conway Morris & Cameron, 2013*) or more recently published contributions (*Conway Morris et al., 2015*; *Maletz & Steiner, 2015*; *Smith, 2015*; *Foster & Gaines, 2016*; *Pates, Daley & Ortega-Hernández, 2017*; *Lerosey-Aubril & Pates, 2018*; *Lerosey-Aubril & Skabelund, 2018*; *Lerosey-Aubril et al., 2018*, *2020*; *Pates, Daley & Lieberman, 2018*; *Pates, Daley & Ortega-Hernández, 2018*; *Conway Morris et al., 2020*; *Lerosey-Aubril, Skabelund & Ortega-Hernández, 2020*) and some new discoveries (R. Lerosey-Aubril, 2019, personal observation). Finally, we complemented the database with information on lifestyle for each taxon. Three broad categories of lifestyles were considered: endobenthic (or infaunal), epibenthic (or epifaunal), and pelagic. Benthic (i.e. bottom-dependent) marine animals are regarded as endobenthic or epibenthic if they spend most of their lives inside the seafloor or on top of it, respectively. An epibenthic lifestyle was also assigned to nektobenthic taxa (sensu *Whalen & Briggs, 2018*)—animals living on the seafloor, but capable of temporary swimming (e.g. most trilobites). We inferred a pelagic lifestyle for animals thought to have spent most of their life within the water column some distance from the seafloor, either as passive drifters (plankton) or active swimmers (nekton).

Fossils were photographed dry or immersed in water, under polarized or cross-polarized illumination, using a Nikon D5500 DSLR fitted with a Nikon 40 mm DX Micro-Nikkor lens or a Canon EOS500D digital SLR Camera fitted with a Canon EF-S 60 mm macro lens. In most cases, images were taken with manual focusing through the focal plane and then stacked using Photoshop CC. Images of counterparts were mirrored to orientate features the same way as in the part, thus facilitating direct comparison between figures/figure elements. Image processing software ImageJ and ImageJ2 were used to make digital measurements (*Schneider, Rasband & Eliceiri, 2012*; *Rueden et al., 2017*). Interpretative drawings and figures were constructed using Photoshop CC and Inkscape 0.92.

The electronic version of this article in Portable Document Format (PDF) will represent a published work according to the International Commission on Zoological Nomenclature (ICZN), and hence the new names contained in the electronic version are effectively published under that Code from the electronic edition alone. This published work and the nomenclatural acts it contains have been registered in ZooBank, the online registration system for the ICZN. The ZooBank LSIDs (Life Science Identifiers) can be resolved and the associated information viewed through any standard web browser by appending the

LSID to the prefix http://zoobank.org/. The LSID for this publication is: urn:lsid:zoobank.org:pub:80914DF2-7D3E-4A02-81DE-156F8E70889E. The online version of this work is archived and available from the following digital repositories: PeerJ, PubMed Central and CLOCKSS.

## Terminology

The terminology used in our descriptions broadly follows *Guo et al. (2019)* and *Lerosey-Aubril et al. (2020)* for frontal appendages. The term 'plate-like endite' is equivalent to 'blade-like endites' (*Guo et al., 2019*), 'broad, elongate endites' (*Moysiuk & Caron, 2019*) and 'elongated ventral spines' (*Daley, Budd & Caron, 2013*) in other recent works. The term 'distal endites' is used to refer to the simple spiniform endites borne on podomeres distal to those which bear plate-like endites, following *Pates et al. (2019)*. This term is equivalent to 'enditic spines' of *Moysiuk & Caron (2019)*. We follow the terminology of *Liu et al. (2020)* for cephalic carapace elements, *Daley & Edgecombe (2014)* for trunk parts, and *Daley & Bergström (2012)* for components of the oral cone, with the addition of the term 'tooth' (used in *Daley, Budd & Caron (2013)* and *Zeng et al. (2018)*) to describe spines protruding from the inner margins of oral cone plates. Additional terminology relating to the orientation and measurements of *Caryosyntrips* frontal appendages follows *Pates & Daley (2017)*. Abbreviations: sag., sagittal; tr., transverse.

## RESULTS

### Systematic Palaeontology

Superphylum PANARTHROPODA *Nielsen, 1995*
Order RADIODONTA *Collins, 1996*
Family HURDIIDAE *Lerosey-Aubril & Pates, 2018*

*Type genus. Hurdia Walcott, 1912* (including *Proboscicaris Rolfe, 1962*).

*Other genera included. Aegirocassis Van Roy, Daley & Briggs, 2015, Buccaspinea* gen. nov., *Cambroraster Moysiuk & Caron, 2019, Cordaticaris Sun, Zeng & Zhao, 2020, Pahvantia Robison & Richards, 1981, Peytoia Walcott, 1911, Stanleycaris Pates, Daley & Ortega-Hernández, 2018, Ursulinacaris Pates, Daley & Butterfield, 2019.* Questionably: *Schinderhannes Kühl, Briggs & Rust, 2009, Zhenghecaris Vannier et al., 2006.*

*Remarks.* The presence of an oral cone made up of plates of different sizes, lightly sclerotized frontal appendages with endites and dorsal spines, and a segmented body covered dorsally by setal structures and bearing triangular lateral flaps allows *Buccaspinea* to be identified as a radiodont. Within Radiodonta, frontal appendages with five or more plate-like endites are only known in members of one family, and so we assign the new taxon to the Hurdiidae.

Originally described as a bivalved euarthropod (*Vannier et al., 2006*), *Zhenghecaris* material was later reinterpreted as central carapace elements of a hurdiid radiodont, following comparisons with new putative hurdiid carapace elements from the Chengjiang
Biota (*Zeng et al., 2018*). The main reasoning included comparisons with other material from the same horizons assigned to a new genus, *Tauricornicaris* (*Zeng et al., 2018*). However, articulated material of *Tauricornicaris latzione*, the type species of this genus, demonstrated the presence of articulated tergites in this animal, which indicates a more crownward position in the euarthropod lineage than radiodonts—*Tauricornicaris* is not a hurdiid (*Cong et al., 2018*). This insight in turn led to uncertainty of the reassignment of *Zhenghecaris* to Hurdiidae. Nevertheless, *Zhenghecaris* was included in a recent phylogenetic analysis aimed at understanding the internal relationships of Radiodonta (*Moysiuk & Caron, 2019*); there, *Zhenghecaris* was interpreted as central carapace elements with posterolateral spinose processes, and its position was resolved as the sister to *Cambroraster*. The precise systematic position of *Zhenghecaris* remains uncertain pending the discovery of more material that supports radiodont affinities, such as an association with lateral carapace elements or frontal appendages, as similar evidence has recently supported the identification of the previously enigmatic *Pahvantia* as a hurdiid radiodont (*Lerosey-Aubril & Pates, 2018*).

The Devonian animal *Schinderhannes* was originally described as a taxon in a crown-ward position relative to Radiodonta, with the support of a phylogenetic analysis (*Kühl, Briggs & Rust, 2009*). More recent studies have supported its identification as a radiodont within a monophyletic Hurdiidae (*Cong et al., 2014*; *Van Roy, Daley & Briggs, 2015*; *Lerosey-Aubril & Pates, 2018*; *Moysiuk & Caron, 2019*). Its frontal appendages show a comparable organization to hurdiids, but the presence of articulated appendages in the trunk constitutes a major departure from radiodont body plans. Detailed information is still lacking on the structure of the oral cone, presence or absence of dorsal flaps, along with finer details of the frontal appendages, body, and tail, all of which could provide additional support for a radiodont/hurdiid affiliation, or alternative assignment, and so it is left as a questionable member of the family pending future redescription.

Genus *Buccaspinea* nov.
urn:lsid:zoobank.org:act:E69418E9-8933-4ABA-ABBB-17FE5540E5F9

*Type species.* *Buccaspinea cooperi* sp. nov., from the Drumian Marjum Formation, Utah, USA.

*Diagnosis.* Hurdiid radiodont exhibiting the following unique combination of characters: oral cone composed of large and small plates bearing large hooked teeth and surrounding a square central opening; frontal appendages attach lateral to oral cone and possess at least 12 podomeres; six unpaired recurved plate-like endites at least five times longer than the podomeres to which they attach, proximal to considerably shorter (one to two times as long as the height of the podomere to which they attach), spiniform distal endites; auxiliary spines on plate-like endites long and robust, projecting distally; at least 11 trunk segments which do not markedly taper posteriorly; setal structures and broad lateral triangular flaps with transverse lines across their width.

*Etymology.* From the Latin *'bucca'* (mouth) and *'spinea'* (spiny, thorny), a reference to the distinctive large oral cone bearing large thorn-like teeth for this new taxon.

*Buccaspinea cooperi* sp. nov.
urn:lsid:zoobank.org:act:80DC43C1-E1A5-4B20-9D7B-D4116122DB85

Figures 2–6

2013 *Hurdia* sp.; *Daley et al.*, p. 35, fig. 24C.
2018a *Hurdia* sp.; *Pates et al.*, p. 104, tab. 1, figs. 2.3 and 2.4.
2020a *Hurdia* sp. nov. A; *Lerosey-Aubril et al.*, pp. 7, figs. 3A and 3B.

*Diagnosis.* As for genus, by monotypy.

*Etymology.* The species name *'cooperi'* honours Jason Cooper, who discovered the specimen and made it available for study.

*Type material, locality, horizon.* The holotype specimen, BPM 1108 (part and counterpart), an almost complete body lacking carapace elements and compressed in oblique-lateral orientation. This specimen was collected in the Drumian strata (*Ptychagnostus punctuosus* Biozone) of the middle Marjum Formation at the Kells Knolls locality (Fig. 1; locality 1 of *Rigby, Church & Anderson (2010)*; GPS: 39.270709, −113.283868) in the House Range, Millard County, Utah.

*Additional specimens.* Three isolated frontal appendages flattened in lateral orientation are tentatively assigned to this new taxon. UU18056.34 was recovered from the slightly older Drumian *Ptychagnostus atavus* Biozone of the upper Wheeler Formation at the 'New Dig Quarry' (GPS: 39.35883333, −113.27861111) in the House Range, Millard County, Utah (*Lerosey-Aubril et al., 2020*). KUMIP 314040 and ROM 59634 originate from the Wuliuan Spence Shale Member (*Ptychagnostus praecurrens* Biozone) of the Langston Formation at the Miners Hollow locality (GPS: 41.6023, −112.0334), Wellsville Mountains, Box Elder County, Utah (*Daley, Budd & Caron, 2013*, fig. 24C; *Pates, Daley & Lieberman, 2018*, fig 2.3, 2.4).

*Description.* Specimen BPM 1108 (a/b) is a near-complete radiodont body composed of an oral cone, paired frontal appendages, trunk segments, lateral flaps and setal structures, which are preserved in dorsolateral view (Figs. 2 and 3). Only the oral cone and frontal appendages are preserved from the head region, and the posterior of the body is not preserved. No eyes, cephalic carapace elements, or internal organs are visible. The specimen measures c. 100 mm (sag.) from the posterior tip of the body to the anterior margin of the oral cone, with the latter structure representing one-quarter of the preserved length (sag.).

    The oral cone (oc, Figs. 2 and 3) is preserved flattened approximately dorsoventrally, and positioned anterior to the trunk. It includes large and small plates (lp, sp, Figs. 4 and 5), all bearing prominent multi-pointed teeth along their inner margins. These teeth measure 1–3 mm from base to tip, being largest at the midpoint of each side of the square

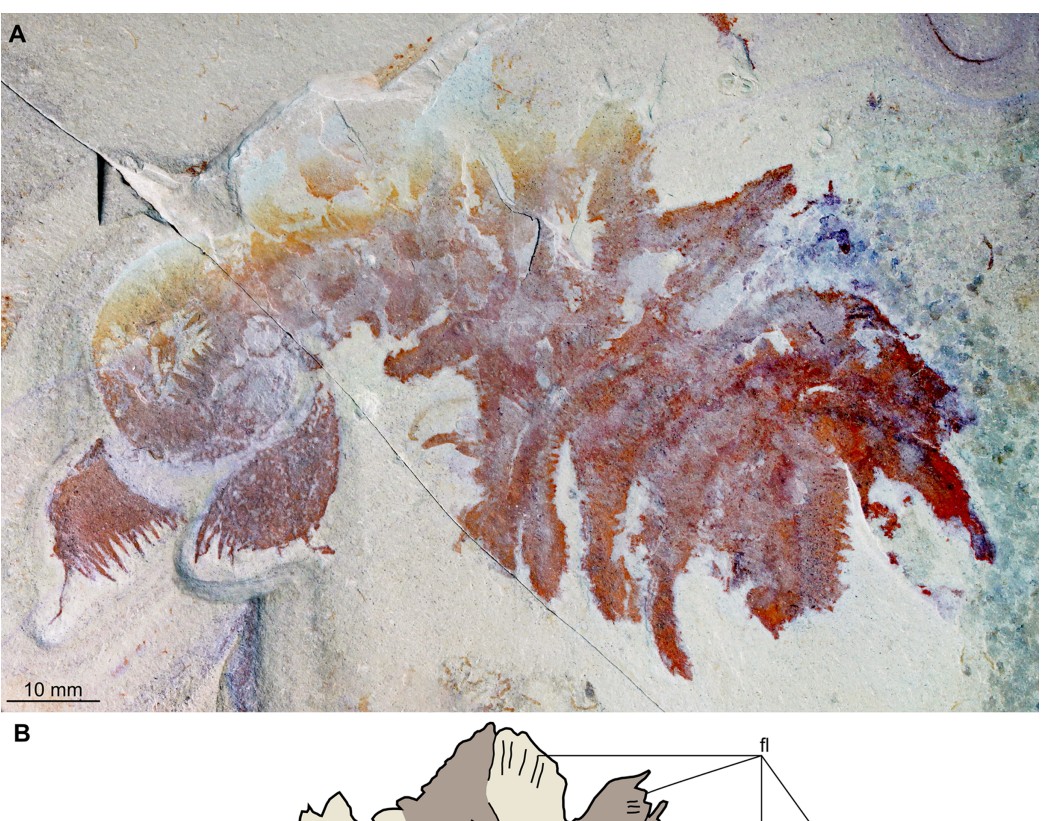

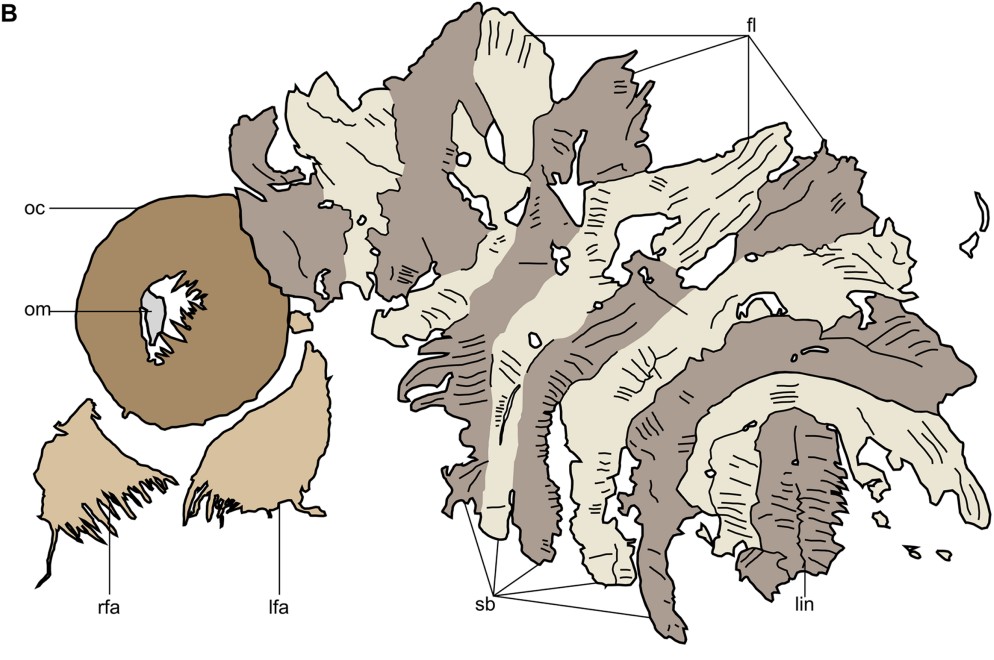

**Figure 2** ***Buccaspinea cooperi* gen. et sp. nov. from the Cambrian (Drumian) Marjum Formation in the House Range of Utah, USA.** (A) Part of holotype specimen (BPM 1108a), general view. (B) Interpretative drawing of (A) (credit: Stephen Pates). Abbreviations: *fl*, triangular lateral flaps; *lfa*, left frontal appendage; *lin*, linear feature on posteriormost preserved body segment; *oc*, oral cone; *om*, organic matter inside central opening of the oral cone; *rfa*, right frontal appendage; *sb*, setal blade.

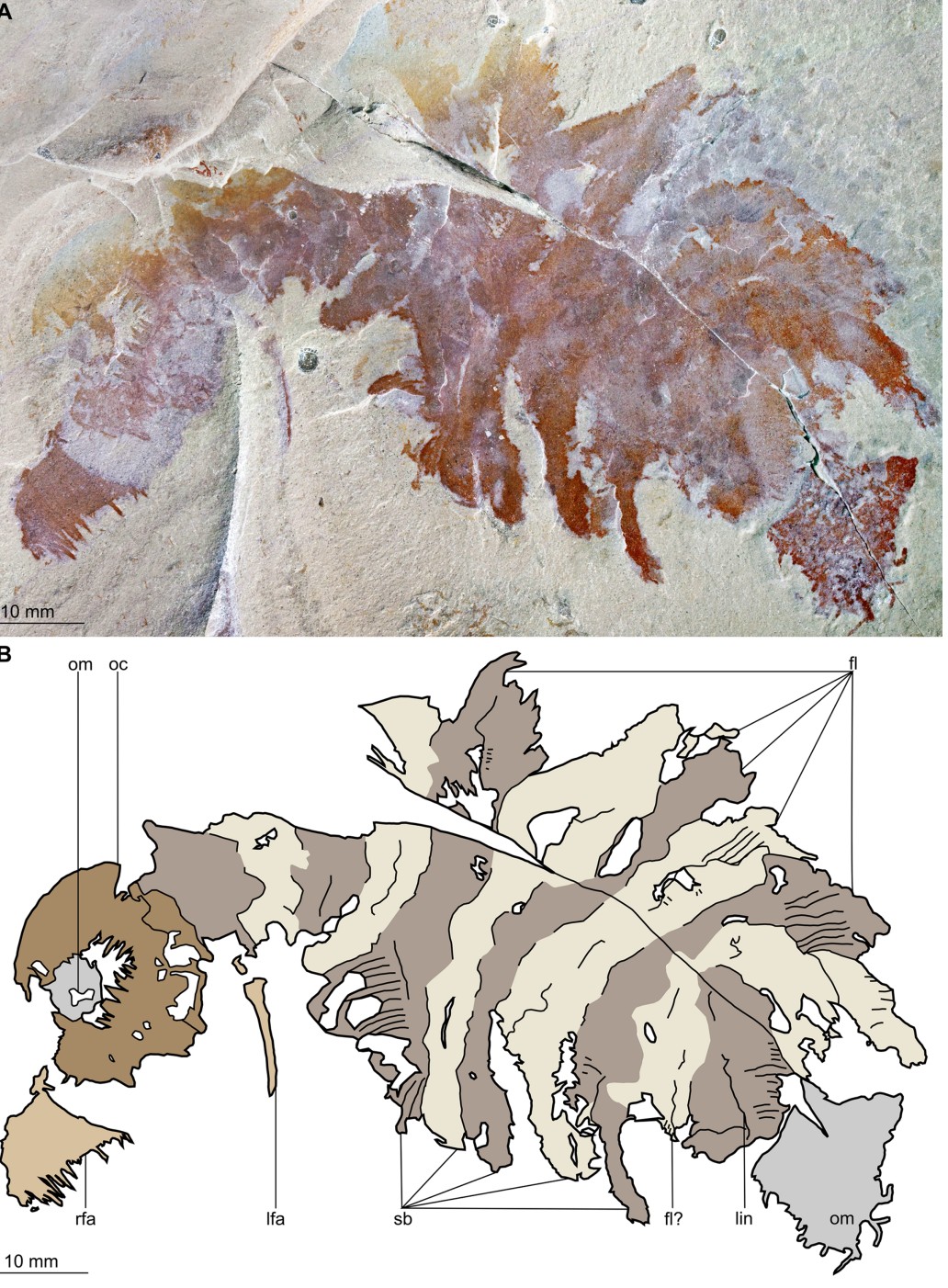

**Figure 3** ***Buccaspinea cooperi* gen. et sp. nov. from the Cambrian (Drumian) Marjum Formation in the House Range of Utah, USA.** (A) Counterpart of holotype specimen (BPM 1108b), general view (mirrored). (B) Interpretative drawing of (A) (credit: Stephen Pates). Abbreviations: *fl*, triangular lateral flaps; *lfa*, left frontal appendage; *lin*, linear feature on posteriormost preserved body segment; *oc*, oral cone; *om*, organic matter inside central opening of the oral cone; *rfa*, right frontal appendage; *sb*, setal blade.

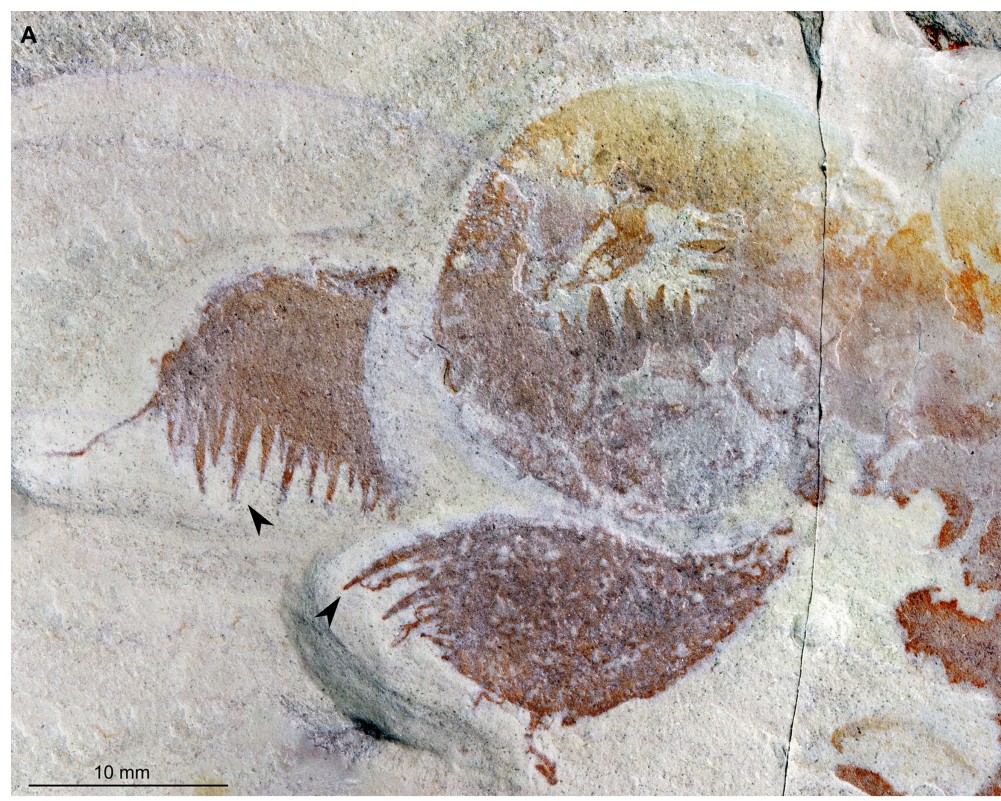

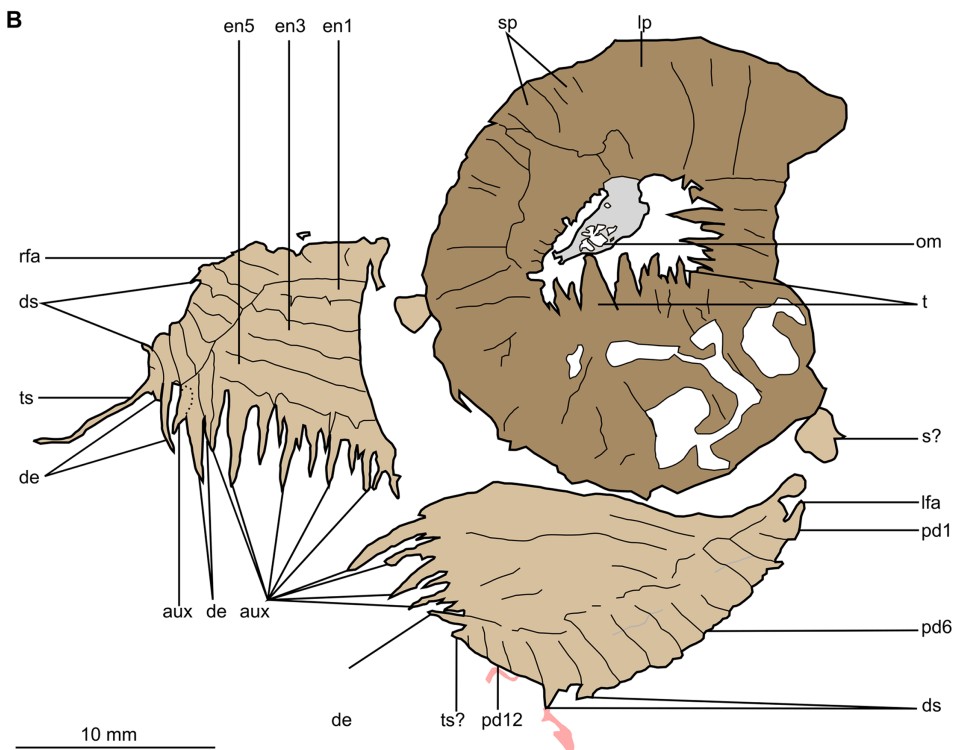

**Figure 4 *Buccaspinea cooperi* gen. et sp. nov. from the Cambrian (Drumian) Marjum Formation in the House Range of Utah, USA.** (A) Part of holotype specimen (BPM 1108a), detailed view of the oral cone and frontal appendages; black arrows indicate longest auxiliary spines used for measurements in the main text. (B) Interpretative drawing of (A) (credit: Stephen Pates). Abbreviations: *aux*, auxiliary spines;

**Figure 4** (continued)
*de*, distal endite; *ds*, dorsal spine; *en*, plate-like endite; *lfa*, left frontal appendage; *lp*, large plate in oral cone; *oc*, oral cone; *om*, organic matter inside central opening of the oral cone; *pd*, podomere; *rfa*, right frontal appendage; *s*, shaft podomere; *sb*, setal blade; *sp*, small plate in oral cone; *t*, teeth on inner margin of oral cone; *ts*, terminal spine.               

opening and decreasing in size towards the corners (t, Figs. 4 and 5). Each plate bears a tooth with one, two, or three points. For the three-pointed teeth, a large central point is flanked by two smaller points (e.g. white and black arrows in Fig. 6A). Teeth are only visible on two of four internal margins of the central opening. This is due to the slightly oblique orientation of preservation of the oral cone, which has also been compacted as indicated by the overlapping teeth towards the corner of the square opening. The boundaries between the plates in the oral cone are not clear or consistently preserved enough on either part to allow the precise arrangement (e.g. triradial, tetraradial) and number of large/small plates to be determined. However, the clear corner and two straight sides visible in the bottom right region of the central opening are of typical tetraradial arrangement (Figs. 4 and 5). In the counterpart, one large plate is visible in the centre of the top row, with smaller plates visible towards both top left and top right corners (lp, sp, Fig. 5). An additional structure is present within the main square opening. This structure, which is fragmented and incomplete, abuts the oral cone along the upper margin of the central opening, and is unlikely to be part of the radiodont mouthparts (om, Figs. 2–5).

One frontal appendage is present on each side of the oral cone, with the plate-like endites facing each other (lfa, rfa, Figs. 2 and 3). Both frontal appendages are preserved at a slight oblique angle and exhibit podomere boundaries visible as simple lines. The latter allow the recognition of at least 12 podomeres in the left appendage (pd12, Fig. 4); adjacent to the oral cone, a patch of fossil material is tentatively interpreted as the proximal most part of the shaft region of the appendage (s?, Fig. 4). Only the dorsal margin of the left appendage is visible in the counterpart (lfa, Fig. 5). The total number of podomeres for the right appendage cannot be determined with certainty. At least six large curved, overlapping plate-like endites (en1–6) are delimited by faint lines on each appendage (en, Figs. 4, 5 and 6B; Fig. S1). These endites are incomplete in the right appendage, partly due to preparation work that has revealed the outline of the oral cone; when complete, they become progressively shorter towards the distal region of the appendage and their tips are curved (left frontal appendage, Fig. 4). En6 on both appendages bears robust and elongate distally facing auxiliary spines (aux, Figs. 4 and 5), the longest of which measure 5.5 mm (right frontal appendage) and 6.5 mm (left frontal appendage) (black arrows, Fig. 4A). On the right appendage, some of those spines belong to more proximal endites and protrude from underneath the distalmost endite (Fig. 6B). Large auxiliary spines can be seen towards the tip of all endites for the left appendage, with the exception of the proximal most endite. On en6 of the right appendage (Figs. 4, 5 and 6B), it can be observed that these spines generally decrease in width and length towards the tip of the endite. Distal to en6 on the same appendage, four much shorter, spiniform endites lacking auxiliary spines can be seen, including a particularly short or incomplete distalmost one (de, Figs. 4

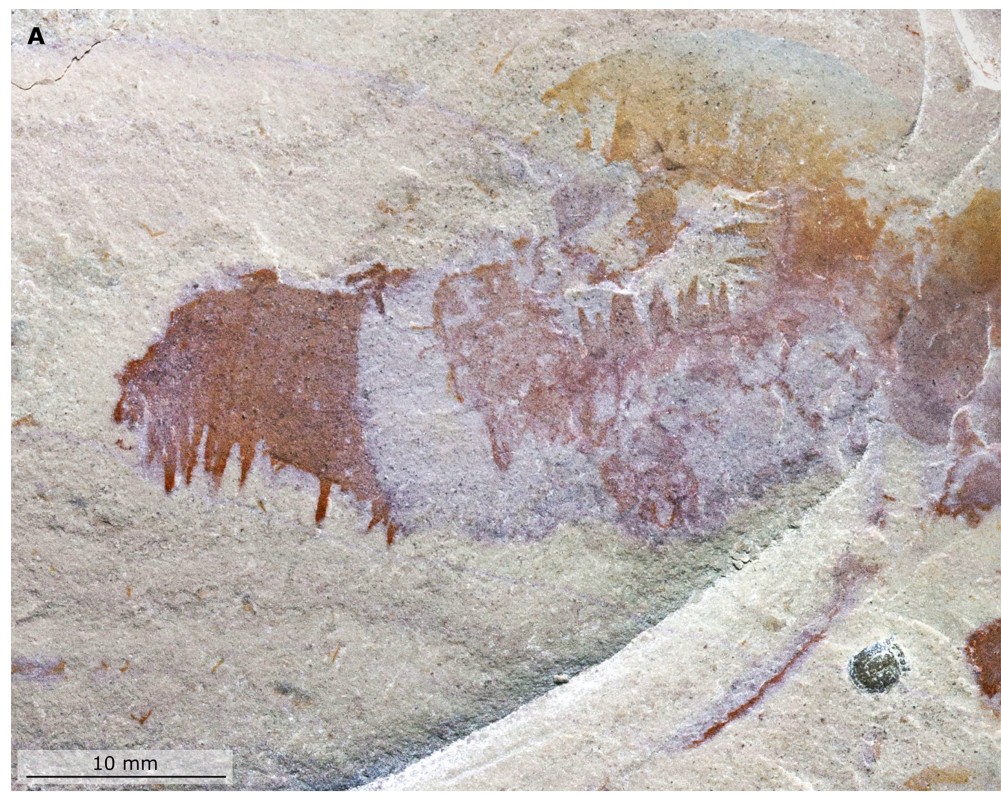

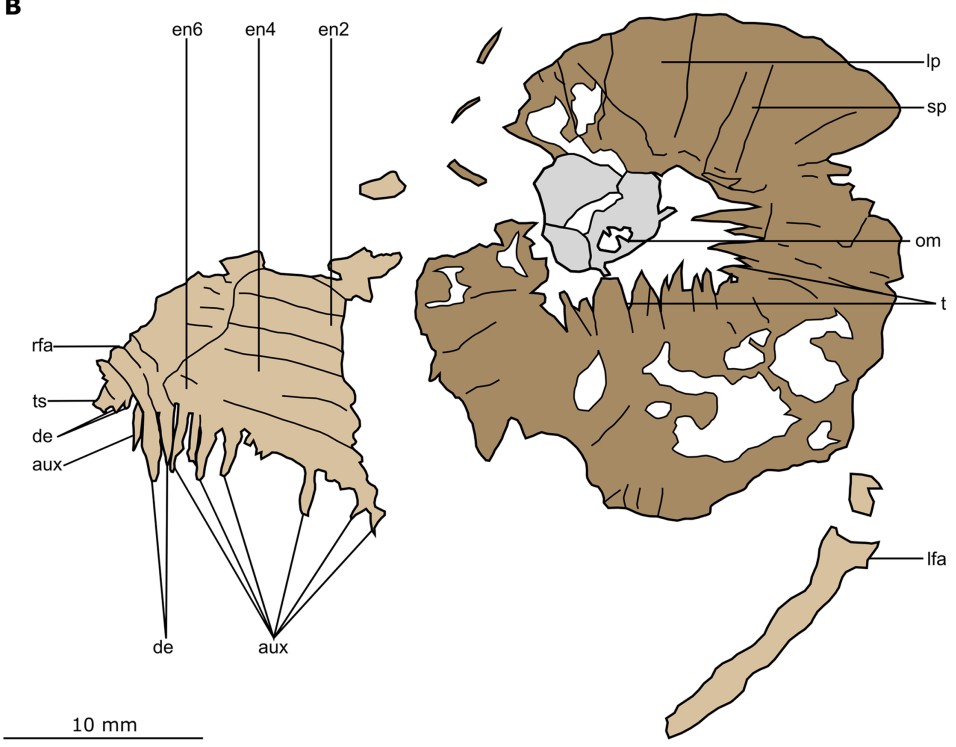

**Figure 5 *Buccaspinea cooperi* gen. et sp. nov. from the Cambrian (Drumian) Marjum Formation in the House Range of Utah, USA.** (A) Counterpart of holotype specimen (BPM 1108b), detailed view of the oral cone and frontal appendages; (B) Interpretative drawing of (A) (credit: Stephen Pates).

**Figure 5 (continued)**
Abbreviations: *aux*, auxiliary spines; *de*, distal endite; *ds*, dorsal spine; *en*, plate-like endite; *ir*, inner row of teeth within oral cone; *lfa*, left frontal appendage; *lp*, large plate; *oc*, oral cone; *om*, organic matter inside central opening of the oral cone; *rfa*, right frontal appendage; *sb*, setal blade; *t*, teeth on inner margin of oral cone; *ts*, terminal spine.

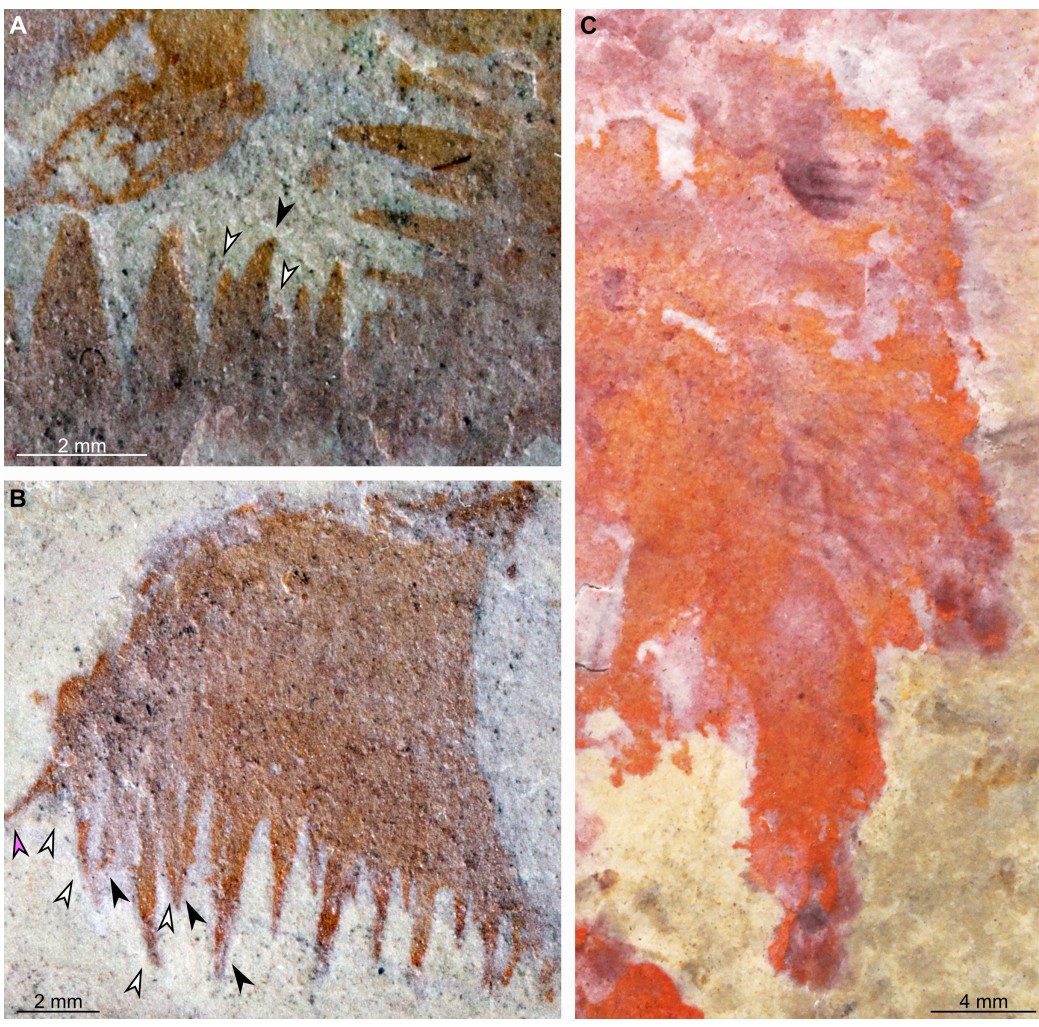

**Figure 6** *Buccaspinea cooperi* **gen. et sp. nov. from the Cambrian (Drumian) Marjum Formation in the House Range of Utah, USA.** (A and B) Part of holotype specimen (BPM 1108a), detailed views of the square opening (A) and right frontal appendage (B). (A) detail of a tricuspid tooth with large central point (black arrow) flanked by two smaller points (white arrows). (B) Shows relationships of terminal spine (pink arrow), auxiliary spines on distalmost plate-like endite (black arrows), and distal endites (white arrows). (C) Counterpart of holotype specimen (BPM 1108b), detailed view (mirrored) of posterior flaps showing transverse lines; photographed under water with cross polarized lighting.

and 5; white arrows, Fig. 6B). Due to the curvature of the appendage, these spiniform endites overlie en6, but they can be distinguished from the similarly-sized auxiliary spines attached to this endite by their orientation: the auxiliary spines are perpendicular to the

endite to which they attach, whereas these distal endites are not (Fig. 6B). A single of these distal endites is clearly visible on the left appendage (de, Figs. 4 and 5). Small triangular dorsal spines are rare but visible on both left and right appendages (ds, Fig. 4) on at least podomeres 5, 9, 10 and 11, suggesting they were likely present along the entire length of the appendage. The base of the terminal spine can be seen on the left appendage (ts, Fig. 4), and the counterpart of the right appendage (ts, Fig. 5). The complete spine is visible only in the right appendage of the part and appears to be long (6.5 mm) and straight, but for a slight kink towards the distal end (ts, Fig. 4). However, it is possible that the morphology of this spine has been obscured by an algal or cyanobacterial filament, because algal or cyanobacterial strands of comparable width can be identified wrapped around dorsal spines of the left appendage (pink strands, Fig. 4), as well as intertwined with some auxiliary spines.

The body, which is incomplete at the posterior, curves towards its left from posterior to anterior and does not appear to taper significantly along its preserved length. The number of segments in the trunk cannot be determined from the central region alone, but counting the flaps and the bands of setal structures suggests that it is composed of 11, possibly 12 of them (alternating color bands, Figs. 2 and 3).

The anterior three segments appear slightly narrower (tr.) than the rest of the body, as suggested by a comparison of the flap size. The lateral flaps are broad and triangular, and they bear transverse lines running parallel to their long axes over their entire surface (Fig. 6C). Lobe-shaped bands of closely packed, linear-shaped blades parallel to one another and perpendicular to the long axis of the bands, can be seen covering the left set of swimming flaps and the dorsal surface of the body (sb, Figs. 2 and 3). The blades change orientation along the length of the band, as shown by the well-preserved organization on the fifth body segment (anteriormost structure labelled 'sb' in Figs. 2 and 3). Interpreted as bands of setal blades, these structures overlie the entire width (tr.) of the bases of flaps five to nine on the right side. At the posterior left side of the trunk, a small triangular structure, tentatively interpreted as a lateral flap, is apparently overlain by a band of setal blades (fl? Fig. 3), but the organization of the different elements of the trunk is obscured by their overlap in this area. An alternative interpretation would be that the band of setal blades and putative flap belong to two distinct segments. The posteriormost preserved structure is trapezoidal in outline and has a linear feature along its midline (lin, Figs. 2 and 3). The linear feature is most likely the centre of a folded band of setal blades. This indicates that the body was disrupted in some way after deposition on the seafloor, in a process that also removed any evidence of a tail fan, lobes, or spines (if present). Its small size compared to the other setal bands suggests that this may be the beginning of the taper at the posterior of the animal, or could potentially represent the posteriormost set of setal blades.

*Remarks.* The new taxon displays a number of similarities to other hurdiids, for example in the morphology of its mouthparts and body characters (Table 1). The unique combination of these features, alongside the frontal appendage characters, warrant the erection of a new genus.

**Table 1 Comparison of the oral cone and trunk morphologies of select hurdiid radiodonts.**

| Anatomical structure | *Aegirocassis* | *Cambroraster* | *Hurdia* | *Peytoia* | *Buccaspinea* |
|---|---|---|---|---|---|
| Oral cone | | | | | |
| Symmetry | Unknown | Tetraradial | Tetraradial | Tetraradial | Tetraradial? |
| Marginal teeth | Unknown | Large, three per plate | Small | Small | Large, hooked, three per plate |
| Inner teeth | Unknown | Present | Present | Absent | Absent |
| Body shape | Oblong | Diamond | Oblong | Diamond | Oblong |
| | (weak to no posterior tapering) | (strong posterior tapering) | (weak to no posterior tapering) | (strong posterior tapering) | (weak to no posterior tapering) |
| Ventral flaps | | | | | |
| Flap-bearing segments[1] | 11 | 8 | 6–9 | 11 | 11+ |
| Lateral flap morphology | Small, triangular With transverse lines across entire width | Small, triangular With transverse lines across entire width | Small, triangular With transverse lines across entire width | Broad, triangular With transverse lines across anterior half | Broad, triangular With transverse lines across entire width |
| Dorsal flaps | Present | Absent | Present | Present | Absent |
| Posterior body region | Unknown | Tailfan, two pairs of caudal lobes | Tailfan, one pair of caudal lobes | No tailfan Trapezoidal termination | Unknown |
| References | *Van Roy, Daley & Briggs, 2015* | *Moysiuk & Caron, 2019* | *Daley et al., 2009*; *Daley, Budd & Caron, 2013*; *Van Roy, Daley & Briggs, 2015* | *Whittington & Briggs, 1985*; *Van Roy, Daley & Briggs, 2015* | This study |

**Note:**
[1] Does not include reduced anterior segments bearing lamellar bands known in *Cambroraster* and *Hurdia*.

A tetraradial arrangement of the large plates, as observed in all other hurdiids where the oral cone is well known (*Cambroraster, Cordaticaris, Hurdia, Peytoia;* Table 1) can be tentatively proposed for *Buccaspinea* based on the locations of the large plates—whether these locations are observed or deduced from the positions of the largest marginal teeth—on the sides of the square-like central opening. The best-preserved large plate can be seen on the upper row of plates in the counterpart (lp, Fig. 5) which is approximately at the centre of that row. As the largest teeth are also found at the centre of the lower and right margins of the cone, these would correspond to the position of the large plate on the upper row, and conform to a tetraradial arrangement for this animal. The teeth in the oral cone of the new taxon are longer and broader relative to the central opening and the size of the oral cone than in *Hurdia, Peytoia*, and to a lesser extent *Cordaticaris* (*Daley, Budd & Caron, 2013*; *Sun, Zeng & Zhao, 2020*), resembling those of *Cambroraster* (*Moysiuk & Caron, 2019*). The plates of the oral cone in *Cambroraster*, the new taxon, and possibly *Cordaticaris* bear multi-pointed teeth, which reduce in size from the midpoint of the side of the central opening towards the corners (*Moysiuk & Caron, 2019*, supplemental figure 6A; *Sun, Zeng & Zhao, 2020*, fig. 6A, B).

The number of flap-bearing trunk segments (not including anterior reduced lamellae or flaps known in *Anomalocaris, Cambroraster, Hurdia* and *Lyrarapax*) described for this new taxon (at least 11) is towards the upper end of what is known in hurdiids (e.g. 11

in *Aegirocassis* and *Peytoia*, eight in *Cambroraster*, six to nine in *Hurdia victoria*), with only *Anomalocaris canadensis* (13) reported as having more among radiodonts (*Whittington & Briggs, 1985*; *Daley et al., 2009*; *Daley, Budd & Caron, 2013*; *Daley & Edgecombe, 2014*; *Van Roy, Daley & Briggs, 2015*; *Moysiuk & Caron, 2019*). The anterior three and posteriormost segments appear to be slightly narrower (tr.) compared to most of the body, but otherwise body segments are of a similar size, as inferred from the relative sizes of the flaps and setal structures. This suggests an approximately oblong outline for the body, similar to what is known in *Aegirocassis* and *Hurdia*, and contrasting with the diamond shape and significant posterior taper of *Cambroraster* and *Peytoia*. It is difficult to draw too many similarities in the organisation of the setal structures with other radiodonts, owing to the slight disarticulation of these features in BPM 1108. Setal structures splay over the dorsal surface and the left side and appear to be dorsal to the triangular swimming flaps. It cannot be determined if *Buccaspinea* has the one-dorsal-block arrangement of *Aegirocassis*, *Cordaticaris* and *Peytoia* (*Whittington & Briggs, 1985*; *Van Roy, Daley & Briggs, 2015*; *Sun, Zeng & Zhao, 2020*), or the alternative arrangement of two separate parallel lateral setal bands known in *Hurdia* (*Daley, Budd & Caron, 2013*). The lateral triangular flaps exhibit transverse lines across the whole width as in *Aegirocassis*, *Cambroraster*, and *Hurdia*, although the flaps in *Buccaspinea* are broader than the flaps of these three hurdiids.

The organization of the frontal appendages in *Buccaspinea* bears many similarities to what is known for other members of Hurdiidae. Most hurdiid frontal appendages consist of a region formed by five or six podomeres that bear plate-like endites, followed by a distal region in which the podomeres have shorter, often spiniform, endites, or no endites at all. In hurdiids with six or more plate-like endites (e.g. *Hurdia*, *Stanleycaris* and a taxon in open nomenclature—?*Peytoia* from the Tulip Beds), the proximal-most is often morphologically distinct from the remaining ones and has been interpreted as belonging to the shaft region (*Pates, Daley & Butterfield, 2019*). The exact morphology of the most proximal of the six endites in this new taxon cannot be determined, but under this hypothesis it would belong to the shaft region, and the remaining five plate-like endites to the distal articulated region. If so, the appendages of this new animal would possess at least 11 podomeres in the distal articulated region, and at least three in the shaft. The recent description of *Cordaticaris*, which exhibits at least eight plate-like endites, suggests that this distinction between shaft and distal articulated region cannot be made on number of endites alone (i.e. hurdiids can have more than five blade-like endites in the distal articulated region), and that a morphological distinction between the shaft endite and endites in the distal articulated region is required to confidently discriminate between these two parts of the appendage. As this cannot be determined for *Buccaspinea*, an alternate interpretation would place all six plate-like endites in the distal articulated region of 12 podomeres, distal to two shaft podomeres lacking endites. Regardless, the endites on the appendages of the new genus reduce slightly in length from proximal to distal, a character that has also been observed in *Hurdia*, and they appear to curve slightly towards the distal portion of the appendage, as seen in *Cambroraster*, *Hurdia*, and *Stanleycaris*. The elongate nature of the plate-like endites in *Buccaspinea*, which are greater than

five times the height of the podomeres to which they attach, is also seen in the filter feeding hurdiids *Aegirocassis* and *Pahvantia*, and to a lesser extent the eudemersal sediment sifter *Cambroraster*, and rare specimens of *Hurdia* (*Daley, Budd & Caron, 2013*; *Van Roy, Daley & Briggs, 2015*; *Lerosey-Aubril & Pates, 2018*; *Moysiuk & Caron, 2019*). The robust and elongate auxiliary spines of *Buccaspinea* bear most similarity to *Cambroraster* and *Hurdia*, and strongly differ from the fine setae of filter feeding hurdiids. It cannot be determined whether these auxiliary spines have hooked tips (*Moysiuk & Caron, 2019*), but their length relative to endite width is more similar to *Cambroraster* (auxiliary spines of a given endite overlapping two or more endites distally) than *Hurdia*. The (absolute) length of the longest auxiliary spines of *Buccaspinea* is comparable to the maximum length reported from *Hurdia* in the Burgess Shale (6 mm *Daley, Budd & Caron, 2013*), and to what is observed in published specimens of *Cambroraster falcatus* (ca. 8 mm, measured digitally from *Moysiuk & Caron, 2019*, fig. 2a). The presence of shorter spiniform endites lacking auxiliary spines in *Buccaspinea* (de, Figs. 4 and 5) is shared with *Cambroraster falcatus* (three), *Hurdia victoria* (one or two), and *Stanleycaris hirpex* (two). *Ursulinacaris grallae* also has two distal podomeres bearing reduced spiniform endites, but the latter are paired as are all endites in this taxon (*Pates, Daley & Butterfield, 2019*). Most hurdiid frontal appendages terminate in a single or pair of short spines, the terminal spine(s), in which the tips are orientated either dorsally (e.g. *Hurdia*) or ventrally (e.g. *Peytoia nathorsti*). One recently described miniature appendage (ca. 2 mm in length) assigned to Hurdiidae from the Ordovician of Wales displays an elongate straight terminal spine, the length of which is approximately a third of that of the appendage (*Pates et al., 2020a*). The terminal spine of this animal also displays a 'U' shaped kink towards its distal end, in the same direction as, but a lower magnitude to, the spine in BPM 1108. These similarities in length (relative to appendage) and shape support the interpretation of the structure protruding from the distal end of the right appendage in *Buccaspinea* as a long terminal spine. In addition, a single specimen of *Caryosyntrips* from the Burgess Shale displays an elongated and apparently flexible projection—albeit thicker than what is observed in *Buccaspinea* and the Welsh hurdiid—at its terminus (*Daley & Budd, 2010*, text-fig 6). On the other hand, algal and cyanobacterial filaments similar in size to this structure are visible around the fossil or associated with prominent parts of it, and therefore a superimposition of a short terminal spine and a single algal or cyanobacterial string cannot be ruled out.

One isolated frontal appendage from the Wheeler Formation, House Range (*Lerosey-Aubril et al., 2020*, fig. 3A. B) and two isolated appendage specimens from the Spence Shale (*Daley, Budd & Caron, 2013*, fig. 24C, D; *Pates, Daley & Lieberman, 2018*, fig. 2.3, 2.4) are tentatively assigned to *Buccaspinea*. All three appendages display characteristics of the plate-like endites strongly reminiscent of BPM 1108. The length and curvature of the endites, alongside the relative length and width of auxiliary spines exceed what is generally observed in *Hurdia*, which is the animal with the most similar frontal appendages. However, all three of these specimens from older Utah Lagerstätten apparently exhibit fewer than three spiniform endites in the distal region. If this could be explained by the poor preservation and the orientation of the distal region in two of

the specimens from the Spence Shale (*Daley, Budd & Caron, 2013*, fig. 24C, D), there are not convincing explanations for the fewer number of distal endites observed in the Wheeler specimen and at least one specimen from the Spence Shale (*Pates, Daley & Lieberman, 2018*, fig 2.3, 2.4), hence the only tentative assignment to the new species. All these specimens possess a short terminal spine; if future findings confirm that the terminal spine of *Buccaspinea* frontal appendages is truly elongate, this would preclude these other isolated frontal appendage specimens from being assigned to the new taxon.

Genus *Pahvantia* *Robison & Richards, 1981*

*Type species*. *Pahvantia hastata* *Robison & Richards, 1981* from the Drumian Wheeler Formation in the House Range of Utah.

*Diagnosis*. See *Lerosey-Aubril & Pates (2018)*.

*Pahvantia hastata* *Robison & Richards, 1981*
Figure 7

*New material*. UMNH.IP6101, 6105, and 6694, complete or near-complete isolated central cephalic carapace elements; precise origins of these specimens unknown, but associated labels mention the Marjum Formation, which crops out in the House Range of western central Utah, USA; exceptionally-preserved fossils have been recovered from the middle part (30–300 m from base) of this formation only, which belongs to the *Bolaspidella* polymerid trilobite Zone and the *Ptychagnostus punctuosus* agnostoid Zone, Drumian Stage, Miaolingian Series.

*Description*. The Marjum central carapace elements hardly differ morphologically from previously illustrated *Pahvantia hastata* specimens, despite being more than four times larger than some (lengths >80 mm, sag; UMNH.IP6105 measures 103 mm; *Pates et al., 2020b*; Table S2). Yet, they allow the recognition of a few morphological details not previously noticed in the taxon. These larger specimens have a slightly wider (tr.) nuchal region relative to the main region of the element (ca. 10 percent increase of the nuchal region width/main region width ratio), when compared to smaller specimens. In addition, UMNH-IP6101 displays two tiny spines on the posterior margin of its left lateral extension (or 'lappet'; Figs. 7B and 7D). One of these marginal spines is located where the line marking the boundary between the lateral extension and the main region meets the margin (Fig. 7D). A restudy of previously published material revealed that this inner marginal spine is preserved in at least four other specimens (KUMIP134187 and 134879, UMNH.IP6088 and 6093; Figs. 7E–7G). In others, no discernible spine occurs, but the margin forms an angle at this point (e.g. KUMIP314089; *Lerosey-Aubril & Pates, 2018*, fig. 1a, b). A second marginal spine is located a short distance abaxially from the first. Its presence could be confirmed in two previously published specimens (KUMIP134187 and 134879; Fig. 7E).

*Remarks*. This is the first report of the presence of *Pahvantia hastata* in the Marjum Formation, this taxon being hitherto only known from the underlying Wheeler Formation

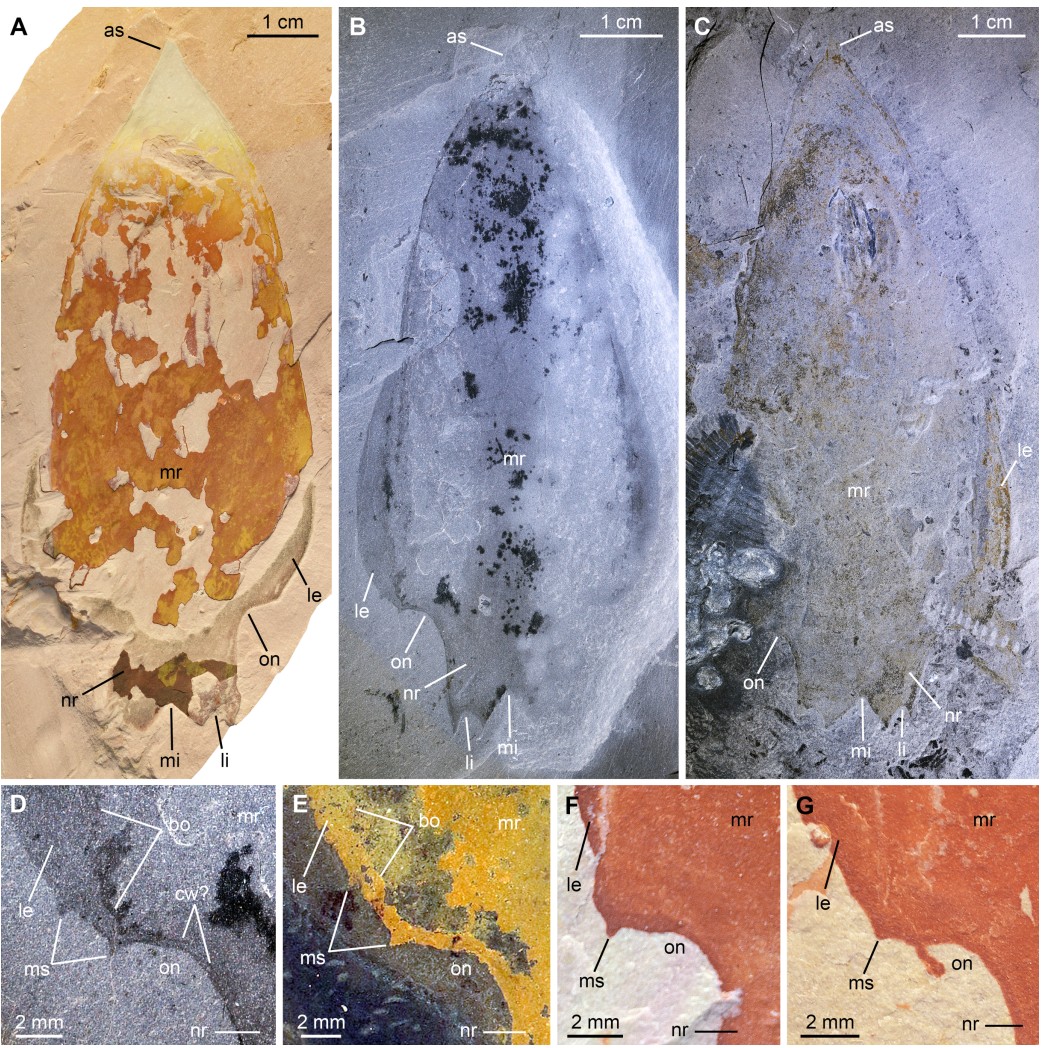

**Figure 7** *Pahvantia hastata Robison & Richards, 1981*, **from the Cambrian (Drumian) Marjum Formation in the House Range of Utah, USA.** All specimens are central carapace elements photographed using cross-polarization, with their anterior ends towards the top. (A) Specimen UMNH.IP6694. (B and D) Specimen UMNH.IP6101, general view (B) and detailed view of the posterior marginal spines (D). (C) Specimen UMNH.IP6105. (E–G) Detailed views of posterior marginal spines in specimens KUMIP134879 (E), UMNH.IP6088 (F), and UMNH.IP6093 (G). Abbreviations: *as*, anterior spine; *bo*, boundary between main region and lateral extension; *cw*, compaction wrinkle; *le*, lateral extension ('lappet'); *li*, lateral indent; *mi*, median indent; *mr*, main region; *ms*, marginal spine; *nr*, nuchal region; *on*, ocular notch.

in both the House Range and the Drum Mountains (*Robison & Richards, 1981*; *Lerosey-Aubril & Pates, 2018*; *Lerosey-Aubril et al., 2020*). The new fossils represent the youngest occurrence of the species and likely extend its biostratigraphical range to the *P. punctuosus* Zone.

The larger sizes of these specimens—UMNH.IP6105 is the biggest specimen of the species yet discovered at 103 mm (sag.)—are not necessarily indicative of biological differences between the Wheeler and Marjum assemblages, but may simply stem from a human bias (e.g. preference for larger fossils of the collector). The slight increase of the

width of the nuchal region relative to the main region of the central carapace element is the only ontogenetic change detectable in the 17 central carapace elements available for study. Otherwise, the Marjum specimens are strikingly similar to previously described specimens, which confirms that the morphology of this central part of the cephalic carapace was strongly constrained, possibly for functional reasons (*Lerosey-Aubril et al., 2020*).

The marginal spines are reminiscent to those projecting along the posterior margins of the posterolateral extensions in *Cambroraster falcatus* (*Moysiuk & Caron, 2019*, figs. 1a, b, g, k, sup. figs. 4C, 5D, 7C). As in *C. falcatus*, marginal spines seem to mark the abaxial limits of the ocular notches, even if the notches in *P. hastata* are represented by concave portions of the margin, rather than actual notches as in *C. falcatus*. Eyes on stalks originating within ocular notches of hurdiid central elements have been described in *Hurdia* (*Daley et al., 2009* fig. 1A, B; *Daley, Budd & Caron, 2013* fig. 3A, B), but no marginal spines were identified in this genus, neither in published material, nor in specimens accessioned at the Museum of Comparative Zoology (*Pates et al., 2020b*, Table S1). The lateral projections of the central element of putative radiodont *Zhenghecaris shankouensis* also bear spines (one each) along their posterior margins, which might be equivalent to those of the two North American taxa, though substantially more robust (*Zeng et al., 2018*, fig. 14A, I). Lastly, *Sun, Zeng & Zhao (2020)* recently described marginal spines in *Cordaticaris* that are similar in number (two per side), location (immediately abaxial to ocular notch), and size (tiny compared to sclerite size) to those of *Pahvantia*, and acknowledged the presence of marginal spines in the latter taxon. These spines represent one of several features of the central carapace element shared by the two taxa (e.g. main region displaying linear pattern, extending into a short anterior spine, and particularly well-differentiated from lateral regions), which suggest a close phylogenetic relationship between them.

Family uncertain
Genus *Caryosyntrips Daley & Budd, 2010*

*Type species. Caryosyntrips serratus Daley & Budd, 2010*, from the Wuliuan Burgess Shale, British Columbia, Canada.

*Diagnosis.* See *Pates & Daley (2017)*.

*Caryosyntrips camurus Pates & Daley, 2017*.
Figures 8, 9

*Material, locality, horizon.* The material consists of two isolated frontal appendages preserved as lateral compressions. Specimen BPM1100, only tentatively assigned to the species, was collected in the Drumian strata (*Ptychagnostus punctuosus* Biozone) of the middle Marjum Formation at the 'Red Wash' locality (locality 716 of *Robison & Babcock, 2011*; GPS: 39.318275°, −113.272793°), House Range, Millard County, Utah. Specimen UMNH.IP 6122 (a, b) was found in the Marjum Formation, and therefore in the House Range of western central Utah, USA, although its exact origins are unknown.

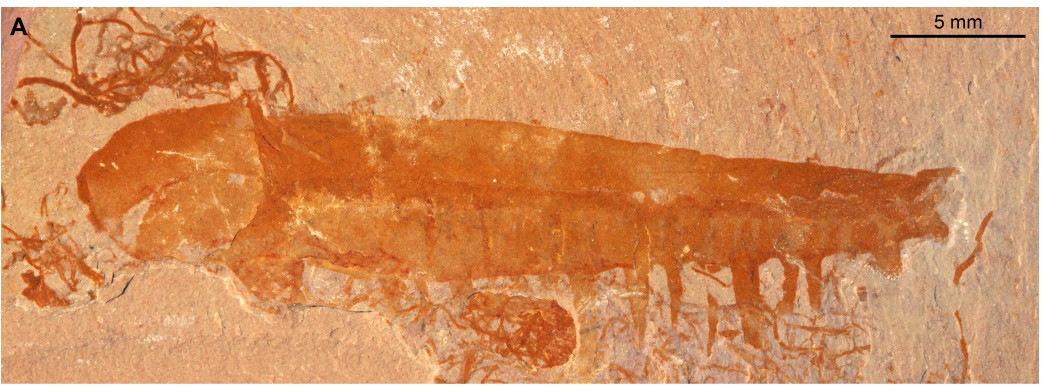

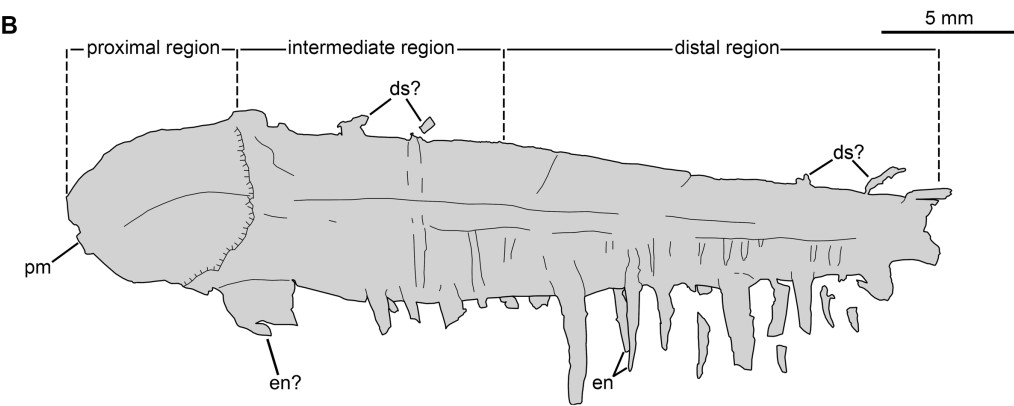

**Figure 8 *Caryosyntrips camurus?* from the Cambrian (Drumian) Marjum Formation in the House Range of Utah, USA.** (A and B) Specimen BPM1100. (A) General view using cross polarized light. (B) Interpretative drawing (credit: Rudy Lerosey-Aubril). Abbreviations: *ds*, projection from dorsal surface, potentially a spine; *en*, paired endites; *pm*, bell-shaped proximal margin.

*Description*. BPM1100 (Fig. 8) is an isolated frontal appendage that measures ca. 33 mm along the dorsal margin. This specimen is composed of a bell-shaped proximal region (Fig. 8), a rectangular intermediate region (ca. 10 mm along dorsal margin), and a trapezoidal distal region (ca. 16 mm along dorsal margin). The proximal region is separated from the rectangular intermediate region by an arcuate boundary (hatched line in Fig. 8B), marking the presence of a second layer of cuticle distally. The dorsal and ventral margins are separated by ca. 4 mm in the rectangular region. A change in slope on the dorsal margin marks the boundary between the proximal and intermediate regions. In the intermediate region the appendage tapers distally at an angle of 12–13° between dorsal and ventral margins until it reaches half of its proximal height at its obliquely truncated tip. A dark coloured band runs at mid-height of the intermediate and distal regions, and the dorsal margin of this band continues as a line that curves ventrally in the proximal region. Ventral to this band are numerous lines running dorso-ventrally, many of which look like proximal extensions of the endites. Some of these structures likely represent the second row of endites, which have been displaced slightly owing to the rotation of the appendage, whereas others may represent incomplete endites or poorly preserved podomere boundaries. Straight endites, which curve slightly towards the

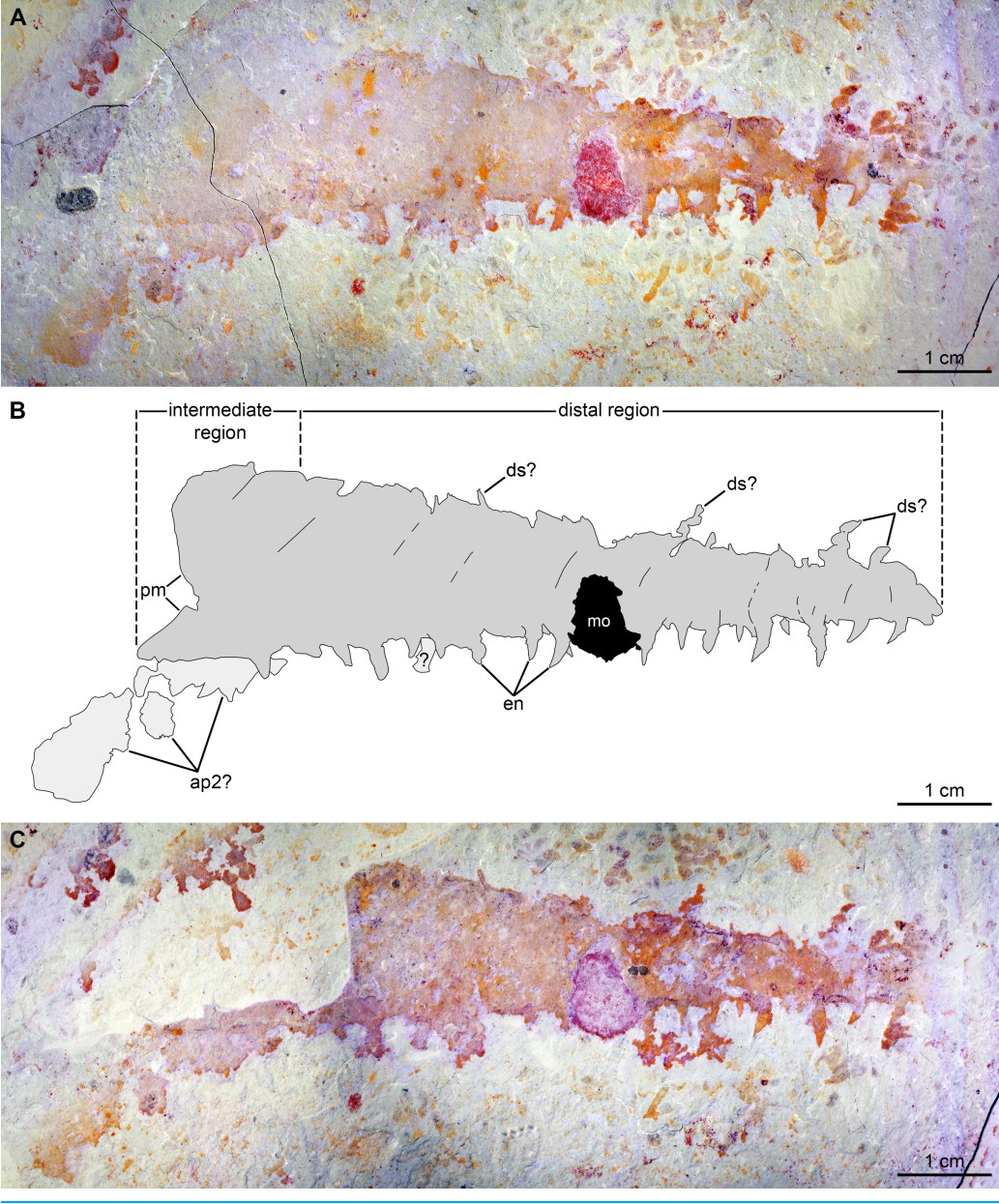

**Figure 9 *Caryosyntrips camurus* from the Cambrian (Drumian) Marjum Formation in the House Range of Utah, USA.** (A–C) Specimen UMNH.IP6122. (A and C) General view of (A) part and (C) counterpart immersed in water (cross polarized light). (B) Composite interpretative drawing, combining details of both part and counterpart (credit: Rudy Lerosey-Aubril). Abbreviations: *ap2*, possible remains of a second appendage; *ds*, projection from dorsal surface, potentially a spine; *en*, endites; *mo*, mineral outgrowth; *pm*, notched proximal margin.

proximal part of the appendage at their distal tips, are closely spaced and attach separately to the ventral portion of the appendage (en, Fig. 8). These endites are of variable length (1.5 to at least 4 mm) and width (0.4 to at least 1.2 mm). The rotation of the appendage cannot account for this variation, as two endites that form a pair (Fig. 8) are of a similar width to one another, despite one being in a deeper plane owing to the rotation

of the appendage. These endites are substantially more slender than the largest and widest endites visible on the appendage.

A 2.5 mm wide structure at the proximal margin of the ventral surface could represent the broken base of an especially large endite, but this structure has a different texture to the other endites. This putative endite could have lost the outer layer of cuticle, or it may simply represent some associated organic matter in the matrix slightly overlain by the *Caryosyntrips* appendage. Similar associated organic matter is abundant in this specimen, with a similar sized block overlying and obscuring some of the endites. The appendage appears smooth and essentially featureless dorsal to the dark colored band. Faint lines running proximo-ventrally from indents of the dorsal margin are interpreted as weakly-expressed podomere boundaries, and four protrusions from the dorsal surface can be seen (ds, Fig. 8). These protrusions could represent poorly preserved dorsal spines, but are more likely additional associated organic fragments, unrelated to the *Caryosyntrips* appendage.

The second specimen, UMNH.IP 6122 (Fig. 9), is an incomplete isolated frontal appendage, which measures ca. 86 mm along the dorsal margin. The specimen is missing the proximal region and parts of the dorsal region. The rectangular intermediate region is bounded proximally by an S-shaped margin, is ca. 17 mm wide (perpendicular distance between dotted lines delineating intermediate region in Fig. 9B) and 20 mm tall (Fig. 9). The trapezoidal distal region measures 69 mm along the dorsal surface, and tapers at an angle of ca. 12° between its dorsal and ventral margins until reaching 3 mm in height at its blunt termination. Faint boundaries separating at least 12 podomeres can be discerned mostly in the dorsal region. Closely spaced endites curve towards the proximal region of the appendage, and vary slightly in length and width (maximum length and width measured are 2 mm and 0.8 mm respectively; en, Fig. 9). At least four projections form the dorsal margin are visible, which potentially represent dorsal spines (ds?, Fig. 9). As these structures are not consistent in terms of their morphology (the proximal-most one is straight, but more distal projections are curved), and abundant organic matter of a similar shape and preservation permeates the matrix, these structures are best interpreted as unrelated to the appendage. Large patches of cuticular material associated with the proximal region, but distinct from it, potentially represent a poorly preserved second appendage (ap2?, Fig. 9).

*Remarks.* These two specimens are assigned to the genus *Caryosyntrips* based on the presence of a subtriangular outline (when flattened), triangular endites, and incomplete podomere boundaries. These two *Caryosyntrips*, the first reported from the Marjum Formation (Drumian), also represent the youngest occurrence of this genus. Three *Caryosyntrips* species were previously reported from older Miaolingian deposits in Laurentia: *C. camurus* (Spence Shale and Burgess Shale, Wuliuan; *Pates & Daley, 2017*), *C. durus* (Wheeler Formation, Drum Mountains; Drumian; *Pates & Daley, 2017*), and *C. serratus* (Burgess Shale and Wheeler Formation, House Range; *Daley & Budd, 2010*; *Pates & Daley, 2017*). The oldest (and largest) member of the genus (*Caryosyntrips* cf. *C. camurus*) is the only known specimen currently described from outside Laurentia (Gondwana, Valdemiedes Formation, Cambrian Stage 4; *Pates & Daley, 2017*) although

the affinities of this specimen have been contested (*Gámez Vintaned & Zhuralev, 2018*; *Pates, Daley & Ortega-Hernández, 2018*).

The three distinct *Caryosyntrips* species are currently defined by the spinosity of their dorsal margins, orientation of endites, and subtle differences in the outline of the appendage (*Pates & Daley, 2017*). The type species, *C. serratus*, bears a row of closely spaced small spines along the dorsal margin, has distally orientated endites, and has a slightly curved dorsal margin. This contrasts with *C. camurus*, which lacks dorsal spines completely, possesses endites which project closer to perpendicular to the ventral margin (except in one specimen from the Spence Shale; *Pates & Daley, 2017*, fig. 4C), and terminates in three podomeres of a very reduced height. The third species, *C. durus*, bears small spines all along its dorsal margin in addition to one large spine per podomere, and endites close to perpendicular to the ventral margin; the entire appendage is triangular in outline, no subrectangular region being differentiated proximally (*Pates & Daley, 2017*). Appendages of *Caryosyntrips* also display a notable amount of intraspecific variation in general outline and endite pattern (shape, position, size, and number) depending on the quality of preservation and orientation of the material, which may considerably complicate assignment to a given species.

The two specimens described here differ in the size, morphology, and spacing of the endites, and the presence/absence of a medial band. The proximal region of *Caryosyntrips* appendages typically display a convex to bell-shaped outline, as observed in BPM 1100 (*Daley & Budd, 2010*, text-fig. 5A; *Pates & Daley, 2017*, figs. 3A, C–F). A concave or sigmoidal proximal margin similar to that of UMNH.IP 6122 has been observed in some *Caryosyntrips* specimens (*Pates & Daley, 2017*, fig. 3B), where the proximal part of the appendage may have broken off along or close to the boundary between two podomeres. Endites projecting approximately perpendicular to the ventral margin as observed in BPM 1100 are known in some specimens of *C. camurus* and *C. durus*, however the substantial size variation of endites is not known in other members of the genus (ratio of endite to appendage is 1:9 in BPM 1100, compared to 1:12 in *C. camurus*; *Pates & Daley, 2017*, fig. 4B, appendix). The endite morphology of UMNH.IP 6122 is most similar to what is seen in the holotype for *C. camurus* (*Pates & Daley, 2017*, fig. 4A). The observation of a medial band running through most of the appendage is another distinctive trait of BPM 1100. A comparable feature occurs in a single previously illustrated specimen of *C. serratus* (*Pates & Daley, 2017*, fig. 3F) but is unknown in any *C. camurus*. A distinction between the ventral portion, with well-expressed podomere boundaries, and an apparently unsegmented dorsal portion is also clearly expressed in the holotype of the same species, but this specimen lacks a clear medial band (*Pates & Daley, 2017*, fig. 3A).

Specimen UMNH.IP 6122 can be confidently assigned to *Caryosyntrips camurus*, assuming that the dorsal projections are taphonomic in origin and do not represent poorly preserved dorsal spines. The shape of the appendage, as well as the size, morphology, and spacing of the endites, all fall within the range of what is known for other members of this species (*Pates & Daley, 2017*). The morphology of the endites is distinct from the only other member of the species from Utah, a partial specimen from the Spence Shale, which displays straight endites with a rounded distal tip (*Pates & Daley, 2017*).
The affinities of specimen BPM 1100 are less clear. Again, assuming that the dorsal projections are taphonomic in origin, the lack of dorsal spines with two rows of simple endites fits with the current diagnosis of *C. camurus*. Whereas the endites of BPM 1100 are larger relative to appendage length than any other member of the species (and genus), this would only require a slight increase in the known morphological variation of spine length within the taxon to accommodate this specimen. However, the unequal spacing of paired endites, and their variation in size, along the ventral margin of the appendage are not observed in any *C. camurus* specimen. These characters (spacing and size variation in endites) may warrant the erection of a new *Caryosyntrips* species in the future, but meanwhile we tentatively assign BPM 1100 to *C. camurus*.

All known species of *Caryosyntrips* are known from Utah Lagerstätten: *C. camurus* in the Spence Shale and Marjum Formation, *C. durus* in the Wheeler strata of the Drum Mountains, *C. serratus* in the Wheeler strata of the House Range), and a potentially novel *Caryosyntrips* species in the Marjum Formation (*Daley & Budd, 2010*; *Pates & Daley, 2017*; *Lerosey-Aubril et al., 2020*; this study).

## DISCUSSION

### Ecological diversity of the Marjum radiodont fauna

The fossils described herein quadruple the known radiodont diversity in the Marjum fauna, adding the taxa *Buccaspinea*, *Caryosyntrips*, and *Pahvantia* to the previously known *Peytoia*. Interestingly, the four Marjum taxa significantly differ from each other in both body and frontal appendage morphologies, which suggests that if they inhabited the waters of the House Range Embayment at the same time, they were probably not ecological competitors (*Daley & Budd, 2010*). *Pahvantia hastata* was recently shown to possess frontal appendages with numerous densely packed setae, structures consistent with suspension feeding habits (*Lerosey-Aubril & Pates, 2018*). Added to an elongate cephalic carapace, this appendicular morphology suggests that *P. hastata* might have inhabited the uppermost layer of the water column, where it fed on micro- to mesoplankton.

*Caryosyntrips* is the least well-known representative of the group—only a partial carapace element is known of its non-appendicular anatomy to date (*Daley & Budd, 2010*). The characteristics of the frontal appendages in this genus, such as their subtriangular outline and incomplete podomere articulations, are so unique among radiodonts that this taxon is typically recovered outside a monophyletic Radiodonta in phylogenetic analyses (*Vinther et al., 2014*; *Cong et al., 2014*; *Van Roy, Daley & Briggs, 2015*; *Liu et al., 2018*; *Lerosey-Aubril & Pates, 2018*; *Moysiuk & Caron, 2019*). *Caryosyntrips* is considered as a free swimmer, similar to other radiodonts and closely related taxa, although its body morphology is unknown and so the extent of its swimming abilities and whether it lived close to the seafloor or high in the water column cannot be determined. It has been speculated that the frontal appendages of *Caryosyntrips* may have worked in a coordinated occlusive motion, with the two appendages moving towards one another to grasp or slice food (*Daley & Budd, 2010*; *Pates, Daley & Ortega-Hernández, 2017*). The size of these appendages (2–20 cm in length; *Pates & Daley, 2017*) and their peculiar inferred function among radiodonts (operating as a pair) both suggest that adult individuals of *Caryosyntrips*

may have fed on much larger items than the micro- to meso-planktonic organisms ingested by *Pahvantia hastata*.

The frontal appendages of *Buccaspinea* bear plate-like endites with extremely robust auxiliary spines, whereas its large oral cone is equipped with particularly robust marginal teeth. The overlap of endites with auxiliary spines would have prevented the capture of prey between endites, and so it is inferred that this animal would have used these endites for sweep feeding, as has been suggested for *Cambroraster* and *Hurdia* which have a comparable frontal appendage organization (*Daley, Budd & Caron, 2013*; *Moysiuk & Caron, 2019*). The large size of the oral cone and robust spines surrounding a square opening strongly suggest that this was used in combination with the appendages for capture and breakdown of prey items, although the exact manner in which the radiodont oral cone functioned is still poorly understood (*Whittington & Briggs, 1985*; *Hagadorn, Schottenfeld & McGowan, 2010*; *Daley & Bergström, 2012*).

The diamond-shaped body of *Peytoia*, with two rows of sub-equal swimming flaps rather long ventral flaps and more reduced dorsal ones (*Whittington & Briggs, 1985*; *Daley, Budd & Caron, 2013*; *Van Roy, Daley & Briggs, 2015*) probably conferred significant swimming power, similar to amplectobeluids and anomalocaridids. The presence of dorsal flaps for steering—and potentially also stability in the water column—in *Peytoia* and other hurdiids (*Van Roy, Daley & Briggs, 2015*), rather than a tail fan and/or caudal rami, suggests reduced agility for this animal when compared to *Anomalocaris* and *Amplectobelua*, whereas the different frontal appendage morphologies of these taxa imply distinct prey handling methods (*Daley & Budd, 2010*).

In summary, the exact autecology of the Marjum radiodonts remains incompletely understood, but there is some evidence that if these taxa truly lived together at a given time, they occupied distinct ecological niches. The positions in the water column that these organisms occupied, their swimming abilities, their sizes at maturity, their feeding mechanics, and the size and origin of the food items they ingested are all factors potentially explaining how these closely-related taxa might have co-occurred in the same ecosystem (Fig. 10). This is similar to the ecological structuring at other Cambrian localities where multiple radiodonts are present, notably at the Burgess Shale where up to seven or more radiodont species have been found at the same site, each interpreted to employ a different feeding strategy, presumably to reduce competition (*Daley & Budd, 2010*).

### Comparison of the Cambrian radiodont faunas from western Utah

The occurrence of four Konservat-Lagerstätten within the Cambrian deposits of the House Range Embayment provides a rare opportunity to study the distribution of different radiodont taxa in both space and time (over ca. 5 million years). Despite the small sample sizes, which are a result of the rare nature of exceptional preservation and significant collecting effort required to obtain radiodont material from these deposits, a comparison of the radiodont faunas of these assemblages (summarized in Table 2) reveals both geographic and temporal signals. Differences are not considered taphonomic in origin as sedimentological studies have demonstrated a similar palaeoenvironmental setting for the Wheeler, Marjum and Weeks Konservat-Lagerstätten in the House Range

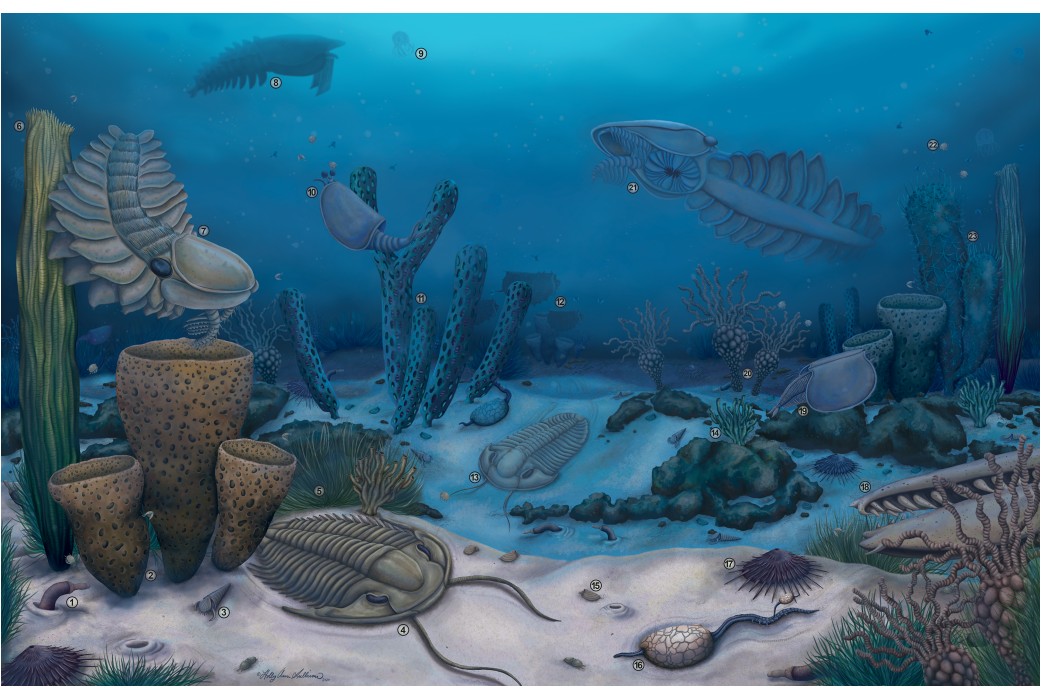

**Figure 10** **Artistic reconstruction of the Cambrian (Drumian) Marjum biota in the House Range of Utah, USA, including radiodont components.** Credit: Holly Sullivan (www.sulscientific.com). Number key for taxa illustrated: 1. *Scathascolex minor*?; 2. *Diagoniella cyathiformis*; 3. *Hyolithes* sp.; 4. *Modocia typicalis*; 5. *Marpolia*-like alga; 6. *Leptomitella metta*; 7. *Peytoia nathorsti*; 8. *Pahvantia hastata*; 9. *Cubozoan* jellyfish; 10. *Perspicaris*? *ellipsopelta*; 11. *Oesia disjuncta/Margaretia dorus*; 12. *Tuzoia guntheri*; 13. *Bathyuriscus fimbriatus*; 14. *Sphenoecium wheelerensis*; 15. *Canthylotreta marjumensis*; 16. *Castericystis vali*; 17. *Choia hindei*; 18. *Caryosyntrips camurus*?; 19. *Branchiocaris pretiosa*?; 20. *Gogia spiralis*; 21. *Buccaspinea cooperi*; 22. *Itagnostus interstrictus*; 23. *Chancelloria* sp.

(*Gaines & Droser, 2005*, *2010*; *Gaines, Kennedy & Droser, 2005*; *Lerosey-Aubril et al., 2018*). In addition, comparable biostratinomic and geochemical contexts have been reconstructed for the intervals hosting exceptional preservation within the Wheeler and Marjum formations (*Gaines & Droser, 2010*). The only notable taphonomic difference between these deposits is the late diagenetic metamorphism that has affected the Weeks Formation, resulting in a different appearance of the soft-bodied fossils recovered from those beds.

The lower Drumian Wheeler radiodont fauna from the House Range (Wheeler-HR) is most diverse, and exemplifies the correlation between taxonomic richness and varied ecological niches within a given assemblage. Taxa with particularly distinctive morphologies are found here (*Lerosey-Aubril et al., 2020*; *Pates, Daley & Ortega-Hernández, 2017*), in addition to all the forms described in the Marjum Formation. An almost complete body specimen of *Anomalocaris* was reported from this locality by *Briggs et al. (2008)*, but restudy suggests that it cannot be accommodated within the type-genus of the family Anomalocarididae (work in progress).

The coeval Wheeler Formation in the Drum Mountains (Wheeler-DM) differs in both species richness and taxonomic composition (Table 2), sharing only *Pahvantia hastata*

 

**Table 2 Taxonomic diversity and fossil richness of the radiodont faunas from the Cambrian Konservat-Lagerstätten of western Utah, USA.**

| Genus | Wheeler-HR | Wheeler-DM | Marjum | Weeks |
|---|---|---|---|---|
| *Amplectobelua* | *Amplectobelua* cf. *A. stephenensis* – 1 FA (*Lerosey-Aubril et al., 2020*) | Absent | Absent | Absent |
| *Anomalocaris* | Absent[1] | Absent[2] | Absent | *Anomalocaris* aff. *A. canadensis* – 5 FA *Anomalocaris* sp. – 1 FA (*Lerosey-Aubril et al., 2014*) |
| *Buccaspinea* | *B. cooperi*? – 1 FA (*Lerosey-Aubril et al., 2020*, their '*Hurdia* sp. nov. A'; This study) | Absent | *B. cooperi* – 1 AB (This study) | Absent |
| *Caryosyntrips* | *C. serratus* – 2 FA (*Pates & Daley, 2017*; *Lerosey-Aubril et al., 2020*) | *C. durus* – 2 FA (*Pates & Daley, 2017*) | *C. camurus* – 2 FA (This study) | Absent |
| *Pahvantia* | *Pa. hastata* – 1 FA, 14 CE, 9 LE (*Robison & Richards, 1981*; *Lerosey-Aubril & Pates, 2018*; *Lerosey-Aubril et al., 2020*) | *Pa. hastata* – 1 CE (*Lerosey-Aubril & Pates, 2018*) | *Pa. hastata* – 3 CE (This study) | Absent |
| *Peytoia* | *Pe. nathorsti* – 1 FA, 3 OC (*Conway Morris & Robison, 1982*; *Pates, Daley & Lieberman, 2018*) | Absent | *Pe. nathorsti* – 1 AB, 1 OC (*Briggs & Robison, 1984*; *Pates, Daley & Lieberman, 2018*) | Absent |
| *Stanleycaris* | *Stanleycaris* sp. – 1 FA (*Pates, Daley & Ortega-Hernández, 2017*) | Absent | Absent | Absent |
| New genus | Absent | Anomalocarididae gen. et sp. nov. – 1 FA (*Halgedahl et al., 2009*) | Absent | Absent |

**Notes:**
Fossil richness is based on published data. Abbreviations used: AB, articulated body (+/− complete), CE, central carapace element, FA, frontal appendage, LE, lateral carapace element, OC, oral cone.

[1] *Briggs et al. (2008)* described an articulated body assigned to *Anomalocaris* from the Wheeler-HR. This specimen is currently under study and is not thought to belong to the genus *Anomalocaris*.

[2] *Robison, Babcock & Gunther (2015)* listed the genus as present in the Wheeler-DM, referring to *Briggs et al. (2008)*. In fact, the Wheeler specimens mentioned in the latter publication were all from the Wheeler-HR.

with the Wheeler-HR fauna. This may support interpretation of *Pahvantia* as a free swimmer predominantly inhabiting the euphotic zone (*Lerosey-Aubril et al., 2018*), or could simply result from its greater abundance and therefore greater chance of being found, as suggested by the numerous carapace elements recovered in the Wheeler-HR (*Lerosey-Aubril et al., 2020*). We include an isolated appendage illustrated by *Halgedahl et al. (2009, fig. 10)* in our tally of radiodonts from Wheeler-DM, with its unique combination of features (e.g. tall podomeres, stout spiniform endites alternating in length) likely representing a new anomalocaridid genus. The different compositions of the two Wheeler radiodont assemblages confirm the view that distinct biotas are preserved in the Wheeler strata of the House Range and the Drum Mountains (*Robison, 1991*; *Robison, Babcock & Gunther, 2015*; *Lerosey-Aubril & Skabelund, 2018*; *Lerosey-Aubril et al., 2020*). However, the Wheeler-DM has to date yielded only four radiodont fossils, and additional discoveries could still significantly change the compositions of the Wheeler radiodont faunas.

The present contribution significantly increases the known diversity of the Marjum radiodont fauna. Although less species rich, this 'middle' Drumian assemblage is strikingly similar to the slightly older ('lower Drumian') Wheeler-HR radiodont fauna (Table 2). *Pahvantia*, *Peytoia* and possibly *Buccaspinea* are represented by the same species in the Wheeler-HR and the Marjum biotas, whereas distinct species of *Caryosyntrips* occur in these two assemblages and the Wheeler-DM. As to the broader palaeobiogeography of these taxa, *Pahvantia* is endemic to western Utah (*Lerosey-Aubril et al., 2020*) and *Buccaspinea* to Utah as a whole (this study), whereas the Laurentian species of *Peytoia* also occurs in the Wuliuan Spence Shale in northern Utah (*Pates, Daley & Lieberman, 2018*) and the Wuliuan Burgess Shale of British Columbia (*Daley, Budd & Caron, 2013*). While *Caryosyntrips durus* has a paleobiogeographic range limited to the Wheeler-DM, *C. serratus* and *C. camurus* are known in both Utah (Wheeler-HR, and Marjum and Spence Shale, respectively) and British Columbia (Burgess Shale) (*Pates & Daley, 2017*). The complexity of these stratigraphical and palaeogeographical distribution patterns, even locally, suggests notable biological and/or ecological differences between radiodont taxa.

Notably, none of the 10 radiodont species recovered from the Wheeler and Marjum Formations has yet been found in the youngest of the Cambrian Lagerstätten of western Utah, the Weeks Formation (Table 2). These Guzhangian strata have yielded two species that are confidently assigned to *Anomalocaris*, even if neither has yet been formally described (*Lerosey-Aubril et al., 2014*). Examination of over 800 exceptionally-preserved fossils collected more recently in the Weeks Formation confirms the presence of two taxa only, their assignment to *Anomalocaris*, and the small size of the individuals inhabiting the House Range Embayment at that time (*Lerosey-Aubril et al., 2014*). This genus is otherwise known from older deposits regionally, in the Cambrian Stage 4 Pioche Formation in eastern Nevada (*Lieberman, 2003*; *Pates et al., 2019*) and the Spence Shale in northern Utah (*Briggs et al., 2008*). As discussed above, the presence of this genus in the Wheeler-HR is doubtful (contra *Briggs et al., 2008*) and therefore, the Weeks specimens are the only fossils confidently assigned to *Anomalocaris* in the Cambrian of western Utah. The absence of hurdiids in the Weeks assemblage is also particularly striking (Table 2), for they are the most common components of the other Miaolingian radiodont faunas of Utah (including the Spence fauna). Thus, radiodonts confirm the singular composition of the Weeks exceptionally-preserved fauna, a uniqueness that was interpreted as evidence for an important biotic turnover around the Drumian/Guzhangian boundary, at least regionally (*Lerosey-Aubril et al., 2018*).

### The Marjum fauna and its pelagic components

The Marjum Formation has yielded 143 species (*Robison, Babcock & Gunther, 2015*), but this whole unit is particularly thick (ca. 430 m) and spans three agnostoid biozones (*Ptychagnostus atavus*, *P. punctuosus*, *Lejopyge laevigata*; *Robison & Babcock, 2011*). Exceptional preservation is confined to the lower part of the *P. punctuosus* Zone only (ca. 30 to 300 m from base; R. Robison, 2019, personal communication), which allows the presently known diversity of this remarkable biota to be quantified at 102 species

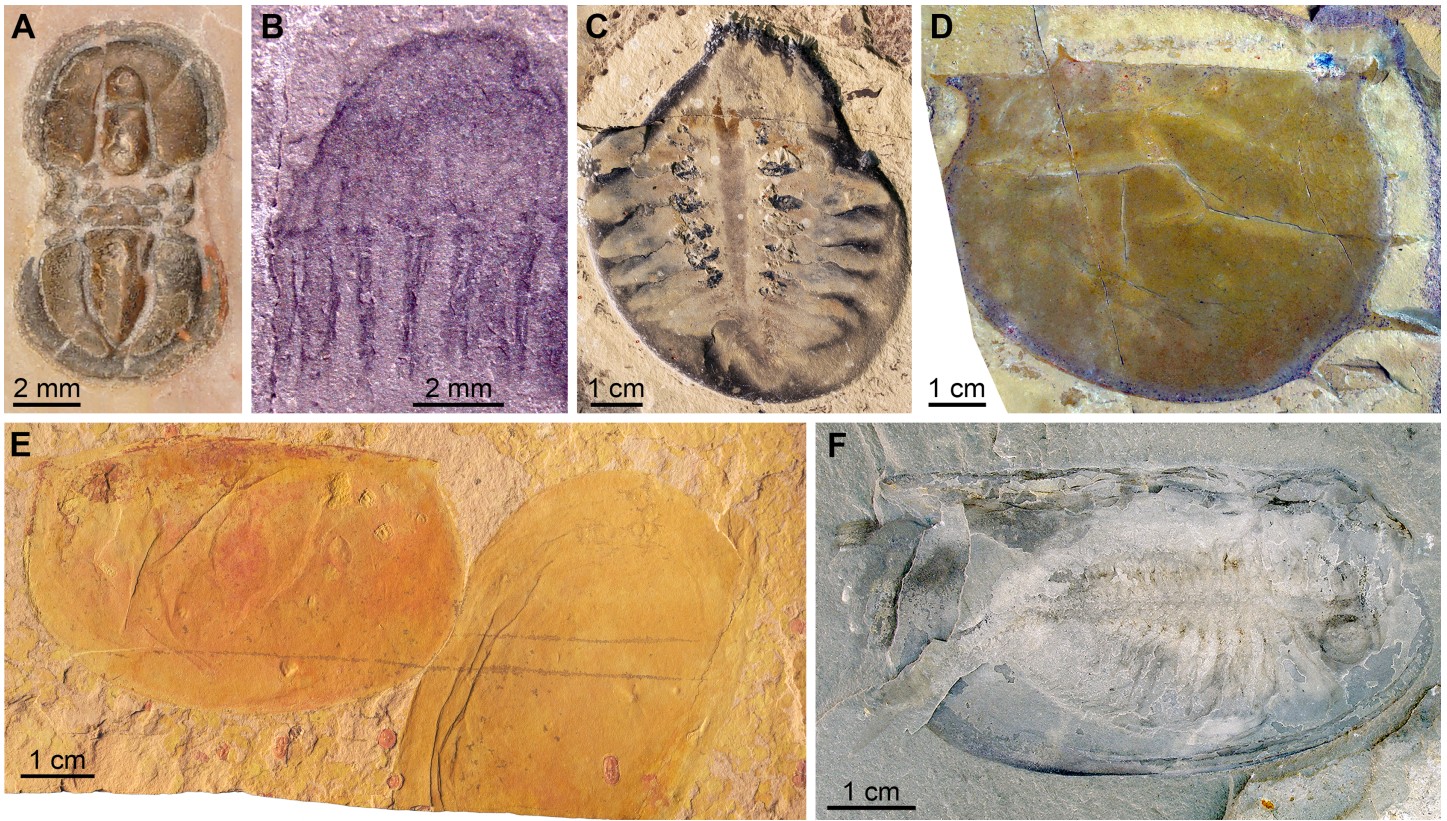

**Figure 11 Examples of pelagic components of the Cambrian (Drumian) Marjum Biota from the House Range of Utah, USA.** (A) Agnostoid *Itagnostus interstrictus*, UMNH.IP5621. (B) Medusiform fossil, UU07021.03 (from *Cartwright et al., 2007*). (C) Radiodont *Peytoia*, USNM. PAL374593. (D) 'Bivalved arthropod' *Tuzoia*, KUMIP153917a (credit: Julien Kimmig). (E) 'Bivalved arthropod' *Perspicaris*, UMNH.IP6323. (F) 'Bivalved arthropod' *Branchiocaris*, KUMIP204797 (credit: Julien Kimmig).

(82 genera), amongst which 97 (77 genera) represent animals (*Pates et al., 2020b*, Data S3). Despite this adjustment for stratigraphic position, the Marjum fauna remains the most diverse of the three exceptionally-preserved Miaolingian assemblages of the House Range, those of the Wheeler-HR (*Ptychagnostus atavus* biozone) and Weeks (*Proagnostus bulbus* biozone) Formations totalling 77 and 81 species, respectively. It also differs from these latter two formations by the noticeably greater proportion of pelagic components (Figs. 11A–11F)—36% of the generic diversity, against 32% in the Wheeler-HR and 17% in the Weeks—a pattern reinforced by the new radiodont occurrences reported herein. This richness in pelagic taxa (see definition above) results in part from a greater diversity of agnostoids (Fig. 11A), which comprise no less than 23 species (14 genera). These small euarthropods, considered herein as forming a clade distinct from trilobites (*Bergström & Hou, 2005*; *Haug, Maas & Waloszek, 2010*; *Edgecombe & Legg, 2013*), account for 18% of the total generic diversity of the Marjum exceptional fauna, which is twice than what they account for in the Wheeler-HR and the Weeks faunas. There are slightly more agnostoid genera than trilobite genera in the middle Marjum assemblage, whereas the diversity of trilobites is more than twice that of agnostoids in the other

two faunas. The Marjum remarkable fauna is also unique in featuring animals that are extremely rare in Cambrian marine assemblages: jellyfish (*Bonino, 2019*). Marjum medusiform fossils (Fig. 11B) were described by *Cartwright et al. (2007)*, who interpreted them as the oldest medusoid representatives of three classes of cnidarians (i.e. Cubozoa, Hydrozoa, and Scyphozoa). Other pelagic components include more common taxa, such as radiodonts—previously only known by the sole genus *Peytoia* (Fig. 11C; *Briggs & Robison, 1984*; *Pates, Daley & Lieberman, 2018*)—and the 'bivalved arthropods' *Branchiocaris*, *Perspicaris*, and *Tuzoia* (Figs. 11D–11F; *Robison & Richards, 1981*; *Briggs & Robison, 1984*).

Interpreting this greater diversity of pelagic components, especially agnostoids, in relation to palaeoenvironmental setting is challenging. Cambrian agnostoids tend to be associated with distal shelf to slope biofacies in low- and mid-latitude regions (*Robison, 1976*; *Sundberg, 1991*; *Pegel, 2000*; *Peng, Babcock & Cooper, 2012*; *Hally & Paterson, 2014*; *Babcock et al., 2015*, *Babcock, Peng & Ahlberg, 2017*), and many taxa have extensive palaeogeographical ranges allowing their use for intercontinental correlation (*Robison, 1976*; *Peng & Robison, 2000*; *Peng, Babcock & Cooper, 2012*; *Álvaro et al., 2013*; *Babcock et al., 2015*, *Babcock, Peng & Ahlberg, 2017*). Whether agnostoids are interpreted as epibenthic (see definition above) or pelagic organisms (*Esteve & Zamora, 2014* and references therein), the doubling of their specific diversity between the upper Wheeler strata and those of the middle Marjum suggests a deepening of the environment. This observation may appear hard to reconcile with the traditional depiction of a relatively continuous filling of the House Range Embayment through deposition of the Wheeler, Marjum, and Weeks Formations (*Miller, Evans & Dattilo, 2012*). This general picture is supported by sequence stratigraphy regionally, which shows that the Marjum Formation records variations of sea levels in the forms of third to fifth order cycles, but overall the evolution of the lithofacies up stratigraphy indicates a general shallowing of the depositional environment (*Smith, 2007*). This shallowing trend is materialized by the southward progradation of shallow platform facies down the carbonate ramp forming the northern margin of the basin (*Rees, 1986*; *Miller, Evans & Dattilo, 2012*). However, sediment accumulation greatly varied within the basin, which was not filled everywhere at the same rate and the same time. Importantly, *Rees (1986)* noted that for most of the existence of the embayment, the low rate of sedimentation in its axial part (e.g. Marjum Pass area) was not sufficient to overcome subsidence, unlike the situation along its northern flank. The inferred relative depth of the central part of the basin somewhat increased during the deposition of the lower (0–200 m) Marjum Formation according to regional sequence stratigraphy (*Smith, 2007*, fig. 30). In other words, while some parts of the basin were being filled, others remained as deep as before or even deepened. Considering that many Marjum localities, including the main sites yielding non-biomineralized fossils (i.e. the Sponge Gully and White Hill localities), correspond to this stratigraphic interval (ca. 30–200 m from base) and geographic area (Marjum Pass and nearby), the observed increase of agnostoid diversity could indeed be interpreted as

supporting evidence for a decoupled bathymetric evolution between the axial and marginal parts of the basin. An alternative, or possibly complementary explanation is that the circulation of water in and out of the embayment changed during the Drumian (e.g. stronger landward currents), allowing enhanced faunistic influences of the oceanic province.

## CONCLUSIONS

The discovery of new material from the Marjum Formation continues to highlight the diversity of Utah Konservat-Lagerstätten, and the description of the new taxon *Buccaspinea cooperi*, known only from Utah deposits, further demonstrates the importance of taking a global approach to our understanding of early animal life. The youngest occurrences of two radiodont genera, *Caryosyntrips* and *Pahvantia*, are also reported from the Drumian Marjum Formation in Utah, which brings a total radiodont diversity of this unit to four taxa, the presence of the youngest *Peytoia nathorsti* in these strata being already well established (*Briggs & Robison, 1984*; *Pates, Daley & Lieberman, 2018*). These radiodont taxa are all known from the younger Wheeler Formation in the House Range, but contrast with the radiodonts of the younger Weeks Formation (Guzhangian), providing further support for a Guzhangian faunal restructuring, at least regionally. These four radiodonts are interpreted as nektonic, and their discovery further documents the relatively high diversity of pelagic animals in the Marjum Formation, in contrast to the other Lagerstätten of the House Range. This may be the result of a slight deepening of this part of the basin during the Drumian, and/or changing ocean circulation at this time bringing in additional pelagic taxa.

## ACKNOWLEDGEMENTS

Richard A. Robison generously provided us with critical data on the stratigraphic positions of the Marjum excavation sites and their fossil assemblages they yield. Carolyn Levitt-Bussian and Randal B. Irmis kindly assisted us during our visits to the Natural History Museum of Utah (UMNH) and facilitated the study of the specimens housed in this institution. These specimens were deposited by the Bureau of Land Management, and we particularly thank Scott E. Foss and Greg McDonald, who also provided curatorial assistance. Manuele Bianchi assisted with the loan of BPM 1108 from the Back to the Past Museum to the University of Lausanne. Peter O. Baumgartner (University of Lausanne) permitted the use of their photography equipment at the University of Lausanne. Paleoartist Holly Sullivan (www.sulscientific.com) helped us to picture how this ecosystem may have looked and created Fig. 10. Julien Kimmig kindly provided the pictures of KUMIP153917a (*Tuzoia*) and KUMIP2047971a (*Branchiocaris*) illustrated in Figs. 11D and 11F, respectively. Bruce S. Lieberman and Julien Kimmig (KUMIP), and Jessica Cundiff and Mark Renczkowski (MCZ), facilitated study of comparative material. The manuscript benefitted from the fruitful comments of L. Babcock, R. LaVine, and an anonymous referee. We would like to express our sincere gratitude to all of these people for their invaluable help before and during the course of our study.

### Funding

Stephen Pates was supported by an Alexander Agassiz Postdoctoral Fellowship awarded by the Museum of Comparative Zoology, Harvard University. The funders had no role in study design, data collection and analysis, decision to publish, or preparation of the manuscript.

### Grant Disclosures

The following grant information was disclosed by the authors:
Museum of Comparative Zoology, Harvard University.

### Competing Interests

The authors declare that they have no competing interests.

### Author Contributions

- Stephen Pates conceived and designed the experiments, performed the experiments, analyzed the data, prepared figures and/or tables, authored or reviewed drafts of the paper, and approved the final draft.
- Rudy Lerosey-Aubril conceived and designed the experiments, performed the experiments, analyzed the data, prepared figures and/or tables, authored or reviewed drafts of the paper, and approved the final draft.
- Allison C. Daley performed the experiments, analyzed the data, authored or reviewed drafts of the paper, and approved the final draft.
- Carlo Kier analyzed the data, authored or reviewed drafts of the paper, and approved the final draft.
- Enrico Bonino analyzed the data, authored or reviewed drafts of the paper, and approved the final draft.
- Javier Ortega-Hernández performed the experiments, analyzed the data, authored or reviewed drafts of the paper, and approved the final draft.

### Data Availability

Supplemental data is available through the Open Science Framework:
Pates, Stephen, Rudy Lerosey-Aubril, Allison C. Daley, Carlo Kier, Enrico Bonino, and Javier Ortega-Hernández. 2020. "Supplementary Data for 'The Diverse Radiodont Fauna from the Marjum Formation of Utah, USA (Cambrian: Drumian).'" OSF. November 16. DOI 10.17605/OSF.IO/Y6XWC.

Fossil specimens figured and studied are deposited in museums and universities, specifically the Back to the Past Museum (BPM), Biodiversity Institute of the University of Kansas (KUMIP), Natural History Museum of Utah (UMNH.IP), Smithsonian National Museum of Natural History (USNM-PAL), University of Utah, Geological Department (UU), and the Museum of Comparative Zoology, Harvard University (MCZ).

Specimen numbers organized by species: *Branchiocaris pretiosa* (KUMIP 204797); *Buccaspinea cooperi* (BPM 1108); *Caryosyntrips camurus* (UMNH.IP 6122); *Caryosyntrips camurus*? (BPM 1100); *Hurdia Victoria* (MCZ 100357, MCZ 103203-103206, MCZ 103208-103212, MCZ 103214-103216, MCZ 103218, MCZ 103297); *Hurdia triangulata* (MCZ 103213, MCZ 103217); *Itagnostus interstrictus* (UMNH.IP 5621); *Pahvantia hastata* (KUMIP 134879, UMNH.IP 6075, UMNH.IP 6088, UMNH.IP 6091, UMNH.IP 6093, UMNH.IP 6095, UMNH.IP 6101, UMNH.IP 6103-6105, UMNH.IP 6118, UMNH.IP 6119, UMNH.IP 6694); *Perspicaris? elliptocephala* (UMNH.IP 6323); *Peytoia nathorsti* (USNM-PAL 374593); medusiform fossil (UU 07021.03).

### New Species Registration

The following information was supplied regarding the registration of a newly described species:

Publication LSID:

urn:lsid:zoobank.org:pub:80914DF2-7D3E-4A02-81DE-156F8E70889E

*Buccaspinea* gen. nov. LSID:

urn:lsid:zoobank.org:act:E69418E9-8933-4ABA-ABBB-17FE5540E5F9

*Buccaspinea cooperi* gen et sp. nov. LSID:

urn:lsid:zoobank.org:act:80DC43C1-E1A5-4B20-9D7B-D4116122DB85.

### Supplemental Information

Supplemental information for this article can be found online at http://dx.doi.org/10.7717/peerj.10509#supplemental-information.

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
