# Peer review of "The diverse radiodont fauna from the Marjum Formation of Utah, USA (Cambrian: Drumian)"

_PeerJ, doi:10.7717/peerj.10509_

## Round 0.1 · original submission · Major Revisions

Dear Dr. Pates and colleagues:

Thanks for submitting your manuscript to PeerJ. I have now received three independent reviews of your work, and as you will see, the reviewers raised some minor concerns about the research. Despite this, these reviewers are optimistic about your work and the potential impact it will have on research studying radiodont fauna from the Marjum Formation. Thus, I encourage you to revise your manuscript, accordingly, taking into account all of the concerns raised by both reviewers.

While the concerns of the reviewers are relatively minor, this is a major revision to ensure that the original reviewers have a chance to evaluate your responses to their concerns. There are many suggestions, which I am sure will greatly improve your manuscript once addressed.

Please note that reviewer 3 has included a marked-up version of your manuscript.

Please use the comments by the reviewers to add missing information where possible (mostly references). Try to restructure your manuscript title for clarity and overall relevance to the work.

Therefore, I am recommending that you revise your manuscript, accordingly, taking into account all of the issues raised by the reviewers. I do believe that your manuscript will be greatly improved once these issues are addressed.

Good luck with your revision,

-joe

·

Basic reporting

No comment.

Experimental design

No comment.

Validity of the findings

No comment.

Additional comments

Pates et al. (2020) provides an exceptional description of new taxa that greatly increase the known diversity of Laurentian radiodonts. I enthusiastically support the acceptance of this paper and am looking forward to it being shared with the greater community. The only thing that I would suggest is to include stratigraphic sections to drive home the facies inference side of the story; however, I see that many of the specimens included cannot be tied to particular sections (let alone particular localities) so the point may be moot.

Reviewer 2 ·

Basic reporting

No comment

Experimental design

No comment

Validity of the findings

Results are supported by evidence but it would be wise to concede that the interpretation of ecology is somewhat speculative.

Additional comments

This is an interesting and important account of the radiodont fauna of the Marjum Formation, one of the major Utah Cambrian Lagerstätten. It includes the description of what is convincingly a new taxon, Buccaspinea, and new reports and information on other radiodonts from the formation. The paper is well written and illustrated. There is a useful comparison of the major soft-bodied Utah assemblages, supported by faunal lists (Supplement) and an extensive discussion of the ecology of the radiodonts which is somewhat speculative (and for that reason I would not lead the title with ecology).

The text and illustrations could be condensed if necessary.

I offer the following suggestions for improving the manuscript (numbers refer to lines).

19 I wouldn’t describe them as common – as top predators, for example, they tend to be rare compared to other elements of the fauna. If you mean radiodont taxa are commonly present that is something different.

20 Seilacher’s concept of Konservat-Lagerstätten extended to well-preserved skeletons. Your definition is too narrow and therefore incorrect. What do you mean by ‘prominence’?

39 You have no evidence for ‘high relative abundance’. A single specimen of Buccaspinea tells you nothing about how rare or common the animal may have been.

50 Konservat Lagerstätten do not equate to strata.

53 What do you mean by ‘equally prolific’ – is diversity similar, fossils similarly abundant …?

81 ‘the latter’ (not these) to make it clear that you are referring to ‘anomalocaridids’.

86 I wonder if the two rami in a paired flap could function indepently to the extent of one contributing propulsion while the other was used for stability and steering.

95 ‘three to five’ presumably out of seven BS genera. Why can’t you be more precise?

113, 147 There’s no merit in giving two estimates for diversity – one more up-to-date than the other.

120 Or it presumably might simply be taphonomic.

149 ‘may be partially explained’ So how does this compare to the stratigraphic spread/duration of fossil occurrences in the other formations?

228 These remarks as a whole refer to genera of Hurdiidae. The first sentence, however, refers specifically to Buccaspinea – that part of the discussion should follow the heading on line 261.

308 ‘are visible from the head region’ is an odd way of indicating that they are the only parts of the head region preserved.

321 ‘The boundaries between the plates in the oral cone are not clear’, but nor are those in the anterior appendage - at least in the illustration. The holotype of Buccaspinea is generously illustrated but what is lacking is a close up view of the anterior appendages in low-angle illumination that would complement the drawing and illustrate some of the detail. Only the overall outline of the appendages is evident in the illustrations.

516 Given the preservation of new morphological details it would be useful to have a reconstruction of the outline of the carapace of Pahvantia.

560 ‘a close phylogenetic relationship’ not relationships.

750 ‘co-occurred in the same ecosystem’ But is there any evidence that they actually co-occurred? Presumably they could have lived in different places and at different times.

756 This section needs to be prefaced by some acknowledgement that the sample size is very small, the animals are not readily preserved, etc. The text could be reduced with reference to Table 2. You don’t have the basis to do more than speculate about ‘the local evolution of radiodont faunas in both space and time’ – you’d need abundance data etc.

850 Here and elsewhere it’s not clear why bivalved arthropods is in parentheses, i.e. ‘bivalved arthropods’. These animals were clearly bivalved and there is nothing to suggest an affinity other than with arthropods.

902 If RAR provided data on stratigraphic positions within the Marjum, why is that information not included – or have I missed it?

Figures 2-6 Every figure legend does not need to repeat all the abbreviations of morphological details.

Figure 6: The photographs do not show the evidence for the morphology of the limbs. You should add a low angle illustration of the anterior appendage to show the breaks of slope that would distinguish the endites, likewise to show that the three cusps of the teeth in the oral cone are not at different levels (i.e. belong to a single tooth). Neither the attitude nor robust nature of the auxialiary spines on the endites are matched in the reconstruction.

Figure 11. Why ‘putative’ hydrozoan? Why ‘bivalved arthropod’ in parentheses? If you don’t agree with these interpretations you should justify your view or provide a citation.

·

Basic reporting

This is an excellent contribution, scientific important, well-done in all respects, and well-written. The language is clear and unambiguous. I think this will stand as a landmark paper in our developing understanding of this group of arthropods.

I have little to add to the paper, or to change, other than some seminal papers or useful reviews that were overlooked or not deemed central to the main points of the paper. These relate to global sea level (which becomes an important part of the argument), biogeographic/paleogeographic patterns, and to agnostoid/'polymeroid trilobite' life habits and classification. The phylogenetic relationships are still in discussion, and citing some of the reviews would be useful.

Experimental design

Excellent. This work is replete with new information, and updates of older data, on a taxonomic group of great importance for understanding the dynamics of the Cambrian evolutionary radiation and the ecology of this group. The research questions are clear, and well researched, with lots of detail provided.

My only concern is with the numerical calculation of swimming vs. non-swimming Cambrian animals. The methodology is not indicated, and it should be. 'Polymerid' trilobites, if they are included as benthic, may be skewing the interpretation toward in the direction of benthic animals. This is unlikely to be true. The trilobites are diverse, and varied ecologic niches are likely. Many were probably nektobenthic. (This is not a major point of the paper, but it is worth a second look, and a clarification of how the information was obtained.)

Validity of the findings

Basically excellent in all respects. My only concern is with the numerical calculation of swimming vs. non-swimming Cambrian animals (noted above). The information relating directly to the radiodont fossils is quite good.

Additional comments

I added comments directly on the ms Word file, and added some key references that should be cited. Please see them.

With quite minor revision, this article is certainly acceptable for publication, and I do not need to see another version.

---

## Round 0.2 · Minor Revisions

Dear Dr. Pates and colleagues:

Thanks for revising your manuscript. The reviewers are very satisfied with your revision (as am I). Great! However, there are a few issues to entertain. Please address these ASAP so we may move towards acceptance of your work.

Best,

-joe

·

Basic reporting

No comment

Experimental design

No comment

Validity of the findings

No comment

Additional comments

The authors have gone above and beyond to address reviewer concerns. I am still very excited to see this research be made available to all, especially to those of us who pursue questions related to the diversity/disparity of stem group arthropods and the evolutionary dynamics of Arthropoda as a whole.

Reviewer 2 ·

Basic reporting

This revised version takes full advantage of the comments of Reviewers 2 and 3 (Reviewer 1 merely provides an endorsement) and should now go forward for publication. I only offer very minor comments for further consideration.

Experimental design

No comment

Validity of the findings

No comment

Additional comments

You provide a robust rejection of the possibility that differences between the Konservat-Lagerstätten might be taphonomic in response to my comment on the last paragraph of the Introduction. You might consider including these observations in the paper.

Retain the comment on Buccaspinea at the beginning of Remarks (under Systematic Paleontology) if you wish, but name Buccaspinea rather than calling it ‘this new taxon’

On line 879 you mean ‘sequence’ not ‘sequential’.

---

## Round 0.3 · accepted · Accept

Dear Dr. Pates and colleagues:

Thanks for revising your manuscript based on the concerns raised by the reviewers. I now believe that your manuscript is suitable for publication. Congratulations! I look forward to seeing this work in print, and I anticipate it being an important resource for groups studying radiodont fauna from the Marjum Formation. Thanks again for choosing PeerJ to publish such important work.

Best,

-joe